# Observing and modelling phytoplankton community structure in the North Sea

David A. Ford[1], Johan van der Molen[2], Kieran Hyder[2], John Bacon[2], Rosa Barciela[1], Veronique Creach[2], Robert McEwan[1], Piet Ruardij[3], and Rodney Forster[2,4]

[1]Met Office, FitzRoy Road, Exeter, EX1 3PB, UK
[2]Centre for Environment, Fisheries and Aquaculture Science (Cefas), Pakefield Road, Lowestoft, NR33 0HT, UK
[3]Royal Netherlands Institute for Sea Research (NIOZ), Department of Coastal Systems and Utrecht University, PO Box 59, 1790 AB Den Burg, Texel, The Netherlands
[4]University of Hull, Cottingham Road, Hull, HU6 7RX, UK

*Correspondence to:* David Ford (david.ford@metoffice.gov.uk)

**Abstract.** Phytoplankton form the base of the marine food chain, and knowledge of phytoplankton community structure is fundamental when assessing marine biodiversity. Policy makers and other users require information on marine biodiversity and other aspects of the marine environment for the North Sea, a highly productive European shelf sea. This information must come from a combination of observations and models, but currently the coastal ocean is greatly under-sampled for phytoplankton data, and outputs of phytoplankton community structure from models are therefore not yet frequently validated. This study presents a novel set of in situ observations of phytoplankton community structure for the North Sea using accessory pigment analysis. The observations allow a good understanding of the patterns of surface phytoplankton biomass and community structure in the North Sea for the observed months of August 2010 and 2011. Two physical-biogeochemical ocean models, the biogeochemical components of which are different variants of the widely-used European Regional Seas Ecosystem Model (ERSEM), were then validated against these and other observations. Both models were a good match for sea surface temperature observations, and a reasonable match for remotely sensed ocean colour observations. However, the two models displayed very different phytoplankton community structures, with one better matching the in situ observations than the other. Nonetheless, both models shared some similarities with the observations in terms of spatial features and inter-annual variability. An initial comparison of the formulations and parameterisations of the two models suggests that diversity between the parameter settings of model phytoplankton functional types, along with formulations which promote a greater sensitivity to changes in light and nutrients, is key to capturing the observed phytoplankton community structure. These findings will help inform future model development, which should be coupled with detailed validation studies, in order to help facilitate the wider application of marine biogeochemical modelling to user and policy needs.

## 1 Introduction

Marine biogeochemical model complexity has long been a subject of debate (e.g. Anderson, 2005). Simpler models require fewer, often better understood parameterisations, but omit processes which are known to be important. More complex models

explicitly include these processes, but require an increased number of tuneable parameters, the ranges of which are often poorly constrained by observations and/or poorly defined where they aggregate over a range of species. No consensus exists within the scientific community, but recent studies have shown that simple to moderately complex models still do the best job of reproducing basic biogeochemical descriptors such as primary production and carbon fluxes (Kwiatkowski et al., 2014; Friedrichs et al., 2007; Ward et al., 2013; Xiao and Friedrichs, 2014). Further studies have suggested that "Models of Intermediate Complexity for Ecosystem assessments" are the most appropriate for fisheries management (Plagányi et al., 2014).

Ultimately, models are tools, and the most appropriate tool should be chosen for the task at hand; different scientific, societal and managerial questions will require models of different complexities. For instance, describing a complex coastal environment will likely require the explicit inclusion of processes which are less important when considering global-scale carbon budgets. Furthermore, some users require more detailed information about the marine environment than simple models can provide, necessitating the use of more complex models.

This demand for detailed information applies to the North Sea, with users and policy makers requiring information about topics including eutrophication and nutrient ratios (Skogen et al., 2014), productivity in relation to fisheries (Chassot et al., 2007), harmful and nuisance algal blooms (Blauw et al., 2010; Kurekin et al., 2014), water clarity (Dupont and Aksnes, 2013; Capuzzo et al., 2015), biodiversity (Brandsma et al., 2013), effects of climate change (van der Molen et al., 2013; Wakelin et al., 2015a), effects of trawling (Allen and Clarke, 2007; van der Molen et al., 2013), and impacts of marine renewable energy generation (van der Molen et al., 2014; van der Molen et al., 2016). In particular, indicators of Good Environmental Status (GES) are required in the context of the Marine Strategy Framework Directive (MSFD; Borja et al., 2013). These include descriptors of food-web structure, trophic status, and biodiversity, and elements of these can be assessed with various modelling approaches (Piroddi et al., 2015; Hyder et al., 2015).

Ecosystem models are central to the delivery of marine ecosystem-based management that is specified in existing legislation (MSFD (EU, 2008), CFP (EU, 2013), WFD (EU, 2000)). These models are important for the design of management measures and to assess the social, economic and environmental performance of management in relation to targets (Defra, 2014; Sutherland et al., 2006). This is done through improving our understanding of the links between pressures (human and environmental) and the response of the system to these pressures. However, ecosystem models are not used frequently in the UK and Europe in support of policy and management (Hyder et al., 2015), despite increasing use in the USA and Australia (Fulton and Link, 2014; Fulton et al., 2007). For models to have a larger impact on policy development and decision-making, modelling approaches need to be more transparent, verifiable, and repeatable than they are at present, as any outputs can be subject to legal challenge (Hyder et al., 2015). Poor uptake of ecosystem models by decision-makers is due to a lack of confidence in and understanding of models. This relates to how models are used, terminology, type of outputs, treatment of uncertainty, required quality standards, and the presentation of model products (Hyder et al., 2015). The use of ecosystem models will become increasingly important as the complexity of marine legislation increases (Boyes and Elliott, 2014). Hence, simple assessment of the skill of models in predicting outcomes (validation - Mackinson, 2014), model comparisons (e.g. Kwiatkowski et al., 2014), and the clear treatment of the uncertainty associated with predictions (Thorpe et al., 2015; Gårdmark et al., 2013; Stewart and Martell, 2015; Tebaldi and Knutti, 2007) are needed to increase the confidence in and uptake of models (Hyder et al., 2015).

At the base of the marine food chain are phytoplankton, and phytoplankton community structure is a fundamental consideration in any assessment of marine biodiversity (Garmendia et al., 2013). Changes in community structure can result from large-scale environmental changes such as temperature rises or eutrophication, with different organisms favouring different conditions. Some organisms that favour changed conditions may be harmful to human health (Roselli and Basset, 2015; Bruggeman, 2009). Alternatively, top-down control by benthic or pelagic grazers can change the size-structure of phytoplankton by selective removal of larger species, resulting for instance in an increased proportion of pico-phytoplankton in areas with dense shellfish aquaculture (Smaal et al., 2013). Phytoplankton vary in size by up to nine orders of magnitude for cell volume (Finkel et al., 2010), with variations in community structure reflected in the size and species of their predators, and the number of links in the food chain (Ryther, 1969; Chavez et al., 2011). Larger cells such as diatoms are consumed directly by copepod grazers, giving a higher transfer of energy and ultimately impacting commercial fish stocks (Jennings and Collingridge, 2015). As the physical structure of the North Sea becomes increasingly well understood due to advances in hydrodynamic modelling (van Leeuwen et al., 2015) and availability of long-term observations (Greenwood et al., 2010; Núñez-Riboni and Akimova, 2015), the potential to predictively model plankton population structure and distribution increases as well.

A common way to model plankton community structure is to take a phytoplankton functional type (PFT) approach, such as is done in variants of the European Regional Seas Ecosystem Model (ERSEM; Baretta et al., 1995). This approach groups phytoplankton into a number of PFTs, based on their general function within the ecosystem (Le Quéré et al., 2005). If information on phytoplankton community structure is to be modelled and provided to users, then it must be validated. Some studies have aimed to validate this against observations (Lewis et al., 2006, Gregg and Casey, 2007; Lewis and Allen, 2009; Hirata et al., 2013), but commonly validation studies go no further than total chlorophyll concentration (Edwards et al., 2012; de Mora et al., 2013). This is largely because there is a lack of observations that contain more detail about community structure against which to compare. Algorithms for deriving phytoplankton community structure from remotely sensed satellite ocean colour observations, either in the form of PFTs or phytoplankton size classes (PSCs) are being developed (Brewin et al., 2011; Brito et al., 2014), but have not yet reached maturity and are not yet widely available to the general scientific community. Moreover, such remote sensing products require a similar level of validation (Brotas et al., 2013). In situ observations are sparse, particularly in shelf seas, and the measured variables may not be easily matched to model outputs, which do not always aggregate neatly over species or size classes.

This study presents a novel set of in situ observations of phytoplankton community structure in the North Sea using accessory pigment analysis (Sherrard et al., 2006), noting that coastal seas are greatly under-represented in the existing global collection of pigment data (Peloquin et al., 2013). Pigment data were analysed so as to give the relative distribution of different size classes, allowing a robust comparison with outputs from ERSEM-type models. Two variants of ERSEM, run by the Centre for Environment, Fisheries and Aquaculture Science (Cefas) and the Met Office, both public bodies in the UK, were validated against these and other observations. The aims of the study were to determine what these new observations add to current scientific understanding of North Sea biogeochemistry, assess the extent to which the models can reproduce the observations, and discuss the implications for current and future user and policy needs, observing strategies and model development.

## 2 Observations

### 2.1 International Bottom Trawl Survey (IBTS)

The International Bottom Trawl Survey (IBTS) is a multi-national ecological research effort established by the International Council for the Exploration of the Sea (ICES) in the early 1970s. Surveys using fisheries research vessels currently take place in the first and third quarter of the year and cover the entire North Sea, using standardised sampling gears and protocols. With cruise lengths of typically 6-8 weeks, each vessel undertakes a gridded survey of the North Sea, repeated each year, in which stations are sampled for groundfish (the primary target of the survey), but also secondary targets such as benthos, seabed litter, and hydrographic parameters. Individual station sampling is often accompanied by visual seabird and cetacean estimates, underway acoustics, and online monitoring of near-surface water quality using FerryBox-type instruments (Petersen et al., 2008). The IBTS thus fits the needs of a multi-disciplinary survey capable of collecting data on human pressures and ecosystem responses for legislation such as the MSFD (http://www.jpi-oceans.eu/multi-use-infrastructure-monitoring). The open data policy of ICES has resulted in many significant publications in fisheries research (Jennings et al., 2002; Daan et al., 2005) and fisheries policy (Rombouts et al., 2013; Shephard et al., 2015).

Prior to 2010, phytoplankton had not been systematically sampled on the UK IBTS. Advances in the autonomous sampling and detection of particles in the water column (e.g. online flow cytometry (Thyssen et al., 2015)), and also the need for high-quality in situ data for validation of satellite remote sensing data, indicated that the addition of phytoplankton to the survey would be beneficial. Hence, sampling of PFTs using high-pressure liquid chromatography (HPLC) - pigment fingerprinting, and analytical flow cytometry (results reported elsewhere) were initiated on the third quarter IBTS cruise of the RV *Cefas Endeavour* in August-September 2010 and subsequent years.

Seawater samples from depths of 4 m ('surface') were collected using 10 l Niskin bottles when weather conditions permitted, or from the ship's bow-intake flow-through clean seawater supply during adverse weather conditions. A known amount of water, typically 1000 ml, was passed through a 200 µm gauze to remove larger zooplankton and debris, then filtered within 1 h on 47 mm GFF filters, which were folded in half, wrapped in aluminium foil and snap frozen in liquid nitrogen dry shippers. On return to shore, samples were transferred to a -80 °C freezer for a storage period of 1-2 months before shipping of samples on dry ice to an accredited HPLC laboratory (DHI Water Quality Institute; Horsholm, Denmark) for chlorophyll-a (chl-a) quantification and full accessory pigment analysis (Schlüter et al., 2011).

Pigment data from the surface stations were quality data controlled in several steps: first, with an initial comparison of HPLC chl-a against independent measures of chlorophyll fluorescence from the fluorometers on the ship's Ferrybox and CTD system. This step corrected a small number of mis-labelled samples. In a second step, anomalies within a sample were detected using methods described by Aiken et al. (2009), e.g. regression of total accessory pigments against chl-a concentration and search for outliers.

Diagnostic pigment analysis was then used on the quality-controlled data set to relate the composition of specific accessory pigments to the relative contribution of different size classes to the total phytoplankton biomass. The designation of specific accessory pigments to algal taxonomic groups of different size, e.g. fucoxanthin and peridinin for large-cell diatoms and di-

noflagellates, has been widely established in the biological oceanographic literature (Uitz et al., 2006, 2008). The equations used to estimate the contribution of pico-phytoplankton (0-2 μm), nano-phytoplankton (2-20 μm) and micro- or net phytoplankton (>20 μm) were later modified by Hirata et al. (2008, 2011) and Brewin et al. (2010). The various methods differ in the degree to which the marker pigments chlorophyll-b (chl-b) and 19-hex-fucoxanthin (19-hex) are attributed to the three

size classes. Here, chl-b and 19-hex were assigned equally to the pico-phytoplankton and nano-phytoplankton size classes. Pico-phytoplankton are therefore represented by zeaxanthin, chl-b and 19-hex; nano-phytoplankton are represented by 19-hex, 19-but, alloxanthin and chl-b; and micro-phytoplankton are represented by fucoxanthin and peridinin. Results are expressed as a proportion of the total chl-a concentration for each station.

## 2.2  Other validation data

As well as the IBTS data introduced in this study, other observation-based products have been used for model validation. Sea surface temperature (SST) has been validated against OSTIA (Operational Sea Surface Temperature and Sea Ice Analysis; Donlon et al., 2012), which is an objective analysis product based on remotely sensed and in situ SST observations. Sea surface chlorophyll and suspended particulate matter (SPM) have been validated against remotely sensed ocean colour products from the Medium Resolution Imaging Spectrometer (MERIS) and Moderate Resolution Imaging Spectroradiometer (MODIS)

sensors, developed by Ifremer using the OC5 algorithm (Gohin et al., 2002, 2005, 2008). Due to the limited availability of observations, nutrient concentrations have been validated against the 1° resolution World Ocean Atlas climatologies (Garcia et al., 2010).

## 3  Models

Two different physical-biogeochemical modelling systems were used in this study: GETM-ERSEM-BFM, run by Cefas, and

NEMO-ERSEM, run by the Met Office. Each is described in turn below, followed by a discussion of their differences and similarities. Existing configurations of each model were used, with no attempt made to increase their similarity. Details of the model configuration and forcing are given in Table 1, and the model domains and bathymetries are shown in Fig. 1.

### 3.1  GETM-ERSEM-BFM

GETM (General Estuarine Transport Model) is a public domain, three-dimensional (3D) finite difference hydrodynamical

model (Burchard and Bolding, 2002; available through http://www.getm.eu). It solves the 3D partial differential equations for conservation of mass, momentum, salt and heat. The ERSEM-BFM (European Regional Seas Ecosystem Model - Biogeochemical Flux Model) version used here is a development of the model ERSEM III (see Baretta et al., 1995; Ruardij and van Raaphorst, 1995; Ruardij et al., 1997; Vichi et al., 2003; Vichi et al., 2004; Ruardij et al., 2005; Vichi et al., 2007; van Leeuwen et al., 2013; van der Molen et al., 2013; van der Molen et al., 2014; van der Molen et al., 2016), and describes the dynamics

of the biogeochemical fluxes within the pelagic and benthic environment. The ERSEM-BFM model simulates the cycles of carbon, nitrogen, phosphorus, silicate and oxygen, and allows for variable internal nutrient ratios inside organisms, based on

external availability and physiological status. The model applies a functional group approach and contains six phytoplankton groups, four zooplankton groups, and five benthic groups, the latter comprising four macrofauna and one meiofauna groups. Pelagic and benthic aerobic and anaerobic bacteria are also included. The pelagic module includes a number of processes in addition to those included in the oceanic version presented by Vichi et al. (2007) to make it suitable for temperate shelf seas:

(i) a parameterisation for diatoms allowing growth in spring, (ii) enhanced transparent exopolymer particles (TEP) excretion by diatoms under nutrient stress, (iii) the associated formation of macro-aggregates consisting of TEP and diatoms, leading to enhanced sinking rates and a sufficient food supply to the benthic system especially in the deeper offshore areas (Engel, 2000), (iv) a *Phaeocystis* functional group for improved simulation of primary production in coastal areas (Peperzak et al., 1998, Ruardij et al., 2005), (v) a new resuspension module for inorganic fine SPM that responds to combined currents and

surface waves, and uses a concentration-dependent settling velocity for improved simulation of the under-water light climate (van der Molen et al., 2017), and (vi) resuspension of particulate organic material, in proportion to the resuspended inorganic SPM and the relative concentrations of organic and fine inorganic matter in the sea bed. The model includes a 3-layer benthic module comprising 53 state variables, which enables it to resolve a high level of detail of benthic processes and benthic-pelagic coupling. New features of the benthic model are: (i) benthic diatoms, and (ii) active oxygen uptake of deposit feeders from

the water column. The first four additional pelagic processes listed above are related, and based on detailed implementation of the dynamic model of phytoplankton growth, explicit chlorophyll content and acclimation of Geider et al. (1997). In nutrient-enriched coastal zones, the competition between and seasonal succession of PFTs is influenced strongly by differences in their photosynthetic capability. The modelled photosynthesis and phytoplankton carbon and chlorophyll content follows Geider et al. (1997) closely, by first calculating light-saturated and nutrient-replete photosynthesis. In the second stage, light-adapted

chlorophyll content is calculated and light limitation and nutrient limitation are applied. These result in changes in the chlorophyll:carbon ratio, and growth. Photo-inhibition is included explicitly in the chlorophyll calculcations, and carbon uptake is calculated before applying nutrient limitation. Under nutrient-limited conditions, diatoms excrete the excess carbon as TEP, which is modelled as a separate state variable. *Phaeocystis* cells, implemented as a simplified version of the model of Ruardij et al. (2005), excrete TEP within the colony. Implicit macro-aggregate sinking rates are calculated as a linear proportion of a

prescribed maximum sinking rate, governed by stickiness rates related to the concentrations of diatoms, TEP and the level of nutrient stress, and also induce sinking of other PFTs. Because (Geider et al., 1997) the initial slope of the P-I curve $\alpha_{chl}$, the light-saturated carbon-specific photosynthesis rate $P_m^C$, the saturation parameter for the growth-irradiance curve $K_I$ and the maximum chlorophyll:carbon ratio $\theta_m$ are related through:

$$\alpha_{chl} = \frac{P_m^C}{\theta_m K_I} \qquad (1)$$

and $K_I$ can be approximated by the light intensity at maximum photosynthetic rate ($K_E$), $K_E$ was used to prescribe light sensitivity. Values were selected to simulate observed successions in the Dutch coastal zone (Table 2).

    The model setup for the North Sea uses a spherical grid with a spatial resolution of approximately 11 km and 25 layers in the vertical (Lenhart et al., 2010; van der Molen et al., 2014, 2015). The model was forced with tidal boundary conditions

from a shelf-scale model, temperature and salinity boundary conditions from a global hindcast (ECMWF-ORAS4; Balmaseda et al., 2013; Mogensen et al., 2012), climatological nutrient boundary conditions, observations-based river run-off and riverine nutrient loads (the National River Flow Archive (data available at http://www.ceh.ac.uk/data/nrfa/index.html) for UK rivers, the Agence de l'eau Loire-Bretagne, Agence de l'eau Seine-Normandie and IFREMER for French rivers, the DONAR database for Netherlands rivers, ARGE Elbe, the Niedersächsisches Landesamt für Ökologie and the Bundesanstalt für Gewässerkunde for German rivers, and the Institute for Marine Research, Bergen, for Norwegian rivers; see also Lenhart et al., 2010), and atmospheric forcing from the ECMWF ERA-40 and operational hindcast (ECMWF, 2006a,b).

## 3.2 NEMO-ERSEM

The hydrodynamic component of the Met Office modelling system is NEMO (Nucleus for European Modelling of the Ocean; Madec, 2008). NEMO is an open source community model originally developed for global ocean modelling (e.g. Storkey et al., 2010), but which has also been recently developed for use in shelf seas (O'Dea et al., 2012). The version used in this study (CO5; O'Dea et al., 2017) is based on NEMO v3.4, and is a development of that described in O'Dea et al. (2012) and Edwards et al. (2012). The main updates from O'Dea et al. (2012) are an upgrade from NEMO v3.2 to v3.4, an increase in vertical resolution from 33 to 51 levels and change of coordinate stretching function, changes to the river inputs and Baltic boundary condition, a change of data assimilation scheme from analysis correction to 3D-Var, and the use of bulk formulae to calculate the input atmospheric fluxes rather than direct forcing. These updates are described further below.

The version of ERSEM used is an alternative development of the original code of Baretta et al. (1995), led by Plymouth Marine Laboratory (PML), and is described in detail by Blackford et al. (2004) and Edwards et al. (2012). The pelagic component includes four phytoplankton and three zooplankton functional groups, and one bacterial group. The benthic component includes aerobic and anaerobic bacteria, suspension feeders, bottom feeders, and the meiobenthos. This version follows the photoacclimation model of Geider et al. (1997) in an adapted form, in which nutrient limitation is applied before the other calculations, leading to much lower estimates of excess carbon, which is excreted as detritus. Photoinhibition is included as an additional parameterisation in the photoacclimation method. SPM is simulated as per Sykes and Barciela (2012).

As part of the Forecasting Ocean Assimilation Model (FOAM; Blockley et al., 2014) suite of models, NEMO-ERSEM is run operationally at the Met Office on a daily basis, providing five-day forecasts of physical and biogeochemical variables for the North-West European Shelf Seas. Analyses and forecasts are publicly available through the Copernicus Marine Environment Monitoring Service (CMEMS; http://marine.copernicus.eu), which is the operational service building on the MyOcean project. Physical and biogeochemical reanalysis products (Wakelin et al., 2015b) are also available through CMEMS, and results from the recent NEMO-ERSEM reanalysis were used in this study.

The model was run on the 7 km resolution Atlantic Meridional Margin (AMM7) domain, covering the entire North-West European Shelf Seas, including the North Sea. There are 51 vertical levels, using a hybrid $\sigma$-S coordinate system with the stretching function of Siddorn and Furner (2013). This uses terrain-following coordinates whilst ensuring a fixed surface resolution of 1 m. Remotely sensed and in situ observations of SST were assimilated using a 3D-Var implementation of the NEMOVAR data assimilation scheme (Waters et al., 2015; O'Dea et al., 2012). River inputs were taken from the E-HYPE

model (Donnelly et al., 2015) for flow values, and from the same climatology as in Edwards et al. (2012) for nutrients and SPM. Lateral boundary conditions for physical variables were taken from a reanalysis of the GloSea5 seasonal forecasting system (MacLachlan et al., 2014) at the Atlantic boundaries, and from the IOW-GETM model (Stips et al., 2004) at the Baltic boundary. For biogeochemistry, lateral boundary conditions for nitrate, phosphate and silicate were taken from the World Ocean
Atlas monthly climatology (Garcia et al., 2010) at the Atlantic boundaries, and zero flux boundary conditions were applied at the Baltic boundary. Zero flux boundary conditions were applied for all other biogeochemical variables at all boundaries. Surface forcing was from the ERA-Interim reanalysis (Dee et al., 2011). The NEMO-ERSEM reanalysis covers the period January 1985 to July 2012, but for practical reasons was run in three sections. The final section, which this study uses, started in November 2003, with physics initial conditions taken from the corresponding date of a non-assimilative hindcast of the entire
reanalysis period. Biogeochemical initial conditions were taken from a winter date of the run of NEMO-ERSEM described in Edwards et al. (2012).

### 3.3   Comparison of the two models

Even though both models used versions of ERSEM, it is reasonable to expect differences in the results. Such differences are inevitably a result of the accumulation of differences between the models. It should be noted that both models were run as usual,
and no attempts were made to increase similarity. It is recognised that this means definitive conclusions cannot be reached here on the exact reasons behind differences in results, and this was not the aim of this study. A preliminary discussion is provided here, with more detailed follow-on experiments proposed in Sect. 5. To help understand the differences in model behaviour, this section summarises the main differences between the two models. We focus on two types of differences: general level differences (Table 1), and differences in phytoplankton parameters and parameterisations (Table 2). For the sake of readability,
and to limit repetition, the following summary is kept at a fairly basic level; for (numerical) detail the reader is referred to the tables.

The two hydrodynamics models were different, and in general used different domains, resolutions, and forcing data. The NEMO-ERSEM model had a larger domain (Fig. 1), at higher resolution, and used more advanced atmospheric forcing. More-over, in NEMO-ERSEM, SST was assimilated, while GETM-ERSEM-BFM had no data assimilation. NEMO-ERSEM's river
runoff originated from a model, that of GETM-ERSEM-BFM from observations. The GETM-ERSEM-BFM model used time series of riverine nutrient inputs whereas the NEMO-ERSEM model used a climatology. The SPM model of NEMO-ERSEM contained explicit size fractions and cohesive interactions, but was only forced by flow velocities, while that of the GETM-ERSEM-BFM model was non-cohesive, with implicit size-related behaviour and included resuspension by both currents and waves (van der Molen et al., 2017). The models also used different initial conditions and spin-up sequences.
Both ERSEM versions share a common origin, both use the same base nutrients (N, P, Si, C), and are both based on a functional type approach. They share four phytoplankton types, three zooplankton types, and a basic bacteria type. Both have a three-layered benthic module, with similar nutrient regeneration mechanisms.

GETM-ERSEM-BFM had a number of additional functional types compared to NEMO-ERSEM: *Phaeocystis* colonies, benthic diatoms, carnivorous zooplankton, filter feeder larvae, epibenthos, benthic predators, and benthic and pelagic nitrifying bacteria. Furthermore, it used a $CO_2$ module, whereas in NEMO-ERSEM this was switched off.

The models used different methods for nutrient affinity, nutrient stress and sinking, and light susceptibility. For nutrient affinity, GETM-ERSEM-BFM used 10-100 times higher affinity values for nutrient uptake. There are two ways to measure phytoplankton nutrient uptake in experiments (Veldhuis and Admiraal, 1987): (i) a short-duration experiment in which nutrients are added to nutrient-deprived algal cultures and uptake rates into the internal nutrient buffer are measured; and (ii) an experiment that lasts a full day in which uptake rates needed for daily growth are measured. The parameters for GETM-ERSEM-BFM were based on short-duration experiments, whereas those for NEMO-ERSEM were based on long-duration experiments. The short-duration parameterisation allows for improved incorporation of the dependencies of cell properties such as cell size and buffer capacity. These features were needed to resolve the competition between diatoms and *Phaeocystis* colonies during excessive spring blooms in the Dutch coastal zone, which terminate through phosphate depletion. In GETM-ERSEM-BFM, nutrient stress of pelagic diatoms leads to excretion of all (new fixated) organic C that cannot be used for growth as carbohydrates (TEP). At high levels of diatoms, this excretion leads to the simulation of the effect of macro-aggregate formation through binding by these carbohydrates, through increases in the sinking rate of live and dead particulate matter. NEMO-ERSEM used a more implicit approach to sinking. For light susceptibility, both models used a photosynthesis-irradiance (P-I) curve approach, but NEMO-ERSEM defined it through the initial slope (alpha), whereas GETM-ERSEM-BFM defined it through the light intensity at maximum photosynthetic rate (Ke). For several elements where both models used the same approach, parameter settings were different: maximum productivity, respiration, excretion, minimum quota for P, lysis, and C:Chl ratios. For these, there was typically more differentiation in settings between phytoplankton functional types in GETM-ERSEM-BFM than in NEMO-ERSEM.

### 3.4 Aggregating model PFTs to match observed PSCs

To allow validation of phytoplankton community structure from the models against the IBTS observations, the four PFTs from NEMO-ERSEM and six PFTs from GETM-ERSEM-BFM must be appropriately aggregated to match the observed PSCs. Diatoms (both models), dinoflagellates (both models) and resuspended benthic diatoms (GETM-ERSEM-BFM only) were considered to be micro-phytoplankton. Flagellates (both models) and *Phaeocystis* colonies (GETM-ERSEM-BFM only) were considered to be nano-phytoplankton. The pico-phytoplankton PFT (both models) was directly mapped to the pico-phytoplankton PSC. For consistency with the IBTS observations, the PFTs and PSCs were expressed as fractions of total chlorophyll concentration, rather than biomass.

## 4 Results

### 4.1 Observations

Each year, the IBTS cruise starts in early August in the Southern Bight of the North Sea off the Thames estuary (51.5° N) and proceeds northwards via a series of longitudinal transects, with each transect taking between one to three days, depending upon the width of the North Sea at each point. The final transect between the Shetland Islands and Norway at 61° N was reached by early September for the 2010 and 2011 IBTS cruises. The spatially-averaged annual mean surface temperature for the North Sea was 9.9 °C in 2010 and 10.0 °C in 2011, which were very close to the long-term average of 10.0 °C for the 1985-2014 period. Hence, the years surveyed represent near-average conditions for temperature.

A continuous recording of chlorophyll fluorescence showed good agreement with the quantity of extracted chl-a determined by HPLC ($r^2 = 0.64$ for 2010 and 0.65 for 2011). The number of same-day match-ups between in situ chl-a and satellite-derived chl-a was low for both years, but a comparison of eight-day averaged surface chl-a from MERIS with in situ data showed an excellent qualitative agreement for both years (Fig. 2). Satellite coverage was more complete in 2011 than 2010. Time series plots and maps of the two cruises showed a number of regularly-occurring features that can be observed at this time of year (labelled 'A' to 'J' on Fig. 2).

A zone of high chl-a ($> 2$ mg m$^{-3}$) was observed with all methods in the coastal waters of Belgium, The Netherlands, Germany and Denmark. This zone extended between points 'A' and 'B' for the map of 2010, and points 'F' and 'H' for 2011. High chlorophyll values ($> 2$ mg m$^{-3}$) were observed in the outer Thames estuary and close to the English east coast as far north as the Humber estuary, but the English coastal zone was not as clearly demarked by high chl-a as the continental coast. The continuous recording of the first 7-10 days of the IBTS thus alternated between moderate chl-a (1 to 2 mg m$^{-3}$) and high chl-a as the vessel covered the southern North Sea between 51.5° N and 55° N. An exceptional bloom event with chl-a of over 6 mg m$^{-3}$ was recorded at location 'G' in 2011, and was clearly visible in MERIS and MODIS images.

The central section of the North Sea between 55° N and 58° N was covered during the second and third weeks of the IBTS. This section showed low chl-a values ($< 1$ mg m$^{-3}$) across most of the zone (Fig. 2), particularly in the region north of 'I', 56.5° N to 58.5° N, 0° E to 3° E, which was a large region with values $< 0.5$ mg m$^{-3}$. To the east, the Danish coastal waters ('B' and 'H') showed high chl-a. The inshore English coast north of the Humber, and Scottish coastal waters, are low in chl-a compared to those further south. A moderate chl-a bloom was evident in the chlorophyll fluorescence trace, MERIS image and extracted chl-a at position 'C' in 2010, and a high chl-a patch was evident close to the Scottish coast at Aberdeen at position 'D'.

The northern North Sea was sampled in weeks three and four (from 28th to 29th August onwards) and was similar in 2010 and 2011. An arc of high chl-a was detected from north of the Scottish mainland through the Orkneys and Shetlands, e.g. from 'D' to 'E' in 2010, with particularly high values at 'E'. In 2011, high values were observed from the Orkneys through to north of the Shetlands at 'J'. The FerryBox chlorophyll fluorescence recorded a further large bloom on 6th September 2011, but this event was not sampled for pigments.

As described in Sect. 2.1, PFTs were determined on the basis of accessory pigment composition. In general, pigment diversity was lower in coastal waters and in the southern North Sea and reached peak diversity in the stratified central North Sea. Fucoxanthin was the dominant accessory marker pigment in the southern North Sea, and 19-hex was dominant in the northern North Sea. Pico-phytoplankton were represented in this analysis by the marker pigments zeaxanthin, chl-b and 19-hex; these

pigments were rare in the southern North Sea below a line from East Anglia to the Wadden Sea, hence pico-phytoplankton contribution was estimated in this region to be less than 10 % of total phytoplankton biomass (Fig. 3). The contribution of pico-phytoplankton increased with increasing latitude so that the area with highest contribution from the smallest PFT was found in both years to be located north of 57° N and east of 0° E. Nano-phytoplankton were represented by the pigments 19-hex, 19-but, alloxanthin and chl-b. The highest percentage contribution of nano-phytoplankton was found in both years to

be located in the central and northern North Sea, including the high chl-a regions around the Shetland and Orkney islands. The largest PFT, micro-phytoplankton, were represented by the pigments fucoxanthin and peridinin. The distribution of this group showed highest abundance in the southern North Sea high chl-a regions near the continental coast, but also in location 'G' (2011) and between 'C' and 'D' in 2010.

The combination of continuous underway logging with autonomous instruments, high precision pigment measurements at

selected stations, and good satellite earth observation coverage allowed the patterns of surface phytoplankton biomass and PFT distribution in the North Sea to be well understood. Together, this provided a solid observational base with which to test biogeochemical model accuracy.

## 4.2   Model validation - domain-scale

This section presents validation of physical and biogeochemical model variables against a range of observation-based products,

in order to assess the models' skill at broader scales than the IBTS observations measured. Detailed validation of phytoplankton community structure against the IBTS observations follows in Sect. 4.3. Since the focus of this study is on the phytoplankton community structure in August 2010 and 2011, most of the validation presented here is for these two months. For more general model validation the reader is referred to Edwards et al. (2012) and Wakelin et al. (2015b) for NEMO-ERSEM, and Lenhart et al. (2010), Aldridge et al. (2012), van Leeuwen et al. (2013), van der Molen et al. (2013), and van der Molen et al., (2016,

2017) for GETM-ERSEM-BFM in various configurations. However, some statistical assessment has been performed here for SST, chlorophyll and SPM over the period March 2010 to October 2011. Two seasons have been defined for this assessment: the growing season and winter. The growing season is defined as March to October, and is averaged over 2010 and 2011. Winter is defined as November 2010 to February 2011. Statistics have been calculated in observation space by performing a bilinear interpolation of the daily mean model fields to the observation locations. Calculations have been performed for

$log_{10}$(chlorophyll) rather than for chlorophyll in order to provide a more Gaussian distribution (Campbell, 1995).

Taylor plots (Taylor, 2001) of SST, $log_{10}$(chlorophyll) and SPM are shown in Fig. 4. SST is a good match for the observations in both the growing season and in winter, although lower correlations are found for both models in August 2010 and 2011 than for the whole seasons. Slightly better statistics are obtained for NEMO-ERSEM than for GETM-ERSEM-BFM, reflecting the assimilation of SST data into NEMO-ERSEM. The statistics for $log_{10}$(chlorophyll) differ more between models and between

seasons, and the models are not as good a match for the observations than with SST, as is common in physical-biogeochemical models. With SPM, the two models show large differences in variability. GETM-ERSEM-BFM has a much higher standard deviation than the observations in both seasons, especially the growing season, whilst the standard deviation of NEMO-ERSEM is too low all year round.

Maps of mean SST for August 2010 and 2011, the months during which most of the IBTS observations were collected, are plotted in Fig. 5, from GETM-ERSEM-BFM, NEMO-ERSEM, and OSTIA. There is a great deal of similarity between NEMO-ERSEM and OSTIA, which is unsurprising since NEMO-ERSEM assimilates SST data, but both NEMO-ERSEM and GETM-ERSEM-BFM are able to simulate the spatial features seen in OSTIA, as well as the inter-annual variability between 2010 and 2011. Boundary effects can be seen in GETM-ERSEM-BFM, which has a smaller domain.

Maps of mean sea surface salinity (SSS) for August 2010 from GETM-ERSEM-BFM and NEMO-ERSEM are plotted in Fig. 6. Overlaid in circles are the in situ SSS observations from the 2010 IBTS cruise. Both models show a good qualitative match for the observations in most regions. The only area which differs substantially between the models is the Norwegian Trench and surrounding area in the north-east North Sea. In GETM-ERSEM-BFM the Norwegian coastal current disperses erroneously, spreading freshwater into the North Sea. This is not seen in the observations, and is an issue of model resolution:

finer-resolution configurations of GETM do not suffer from this. The coarseness of the resolution also accounts for the Rhine freshwater plume being wider than observed.

Maps for sea surface chlorophyll concentration are plotted in Fig. 7, from the two models and the OC5 products. Daily ocean colour coverage is incomplete due to cloud cover, so the observations plotted here are simply a composite of all observations available during the month, rather than a true monthly mean. To ensure a fair comparison, the daily mean model fields were

bilinearly interpolated to observation locations, and equivalent composites plotted rather than the true model mean. Van der Molen et al. (2017) presented a comparison of sub-sampled model results, accounting for cloud cover, of SPM with the true model mean, which suggested noticeable differences in winter, but only small differences in summer. The match between the models and the observations is not as good for chlorophyll as for SST, but both models were still able to capture some of the observed features. In the central and northern North Sea, which has the lowest chlorophyll concentrations, values were generally

under-estimated by GETM-ERSEM-BFM and over-estimated by NEMO-ERSEM. High coastal chlorophyll values were better simulated by GETM-ERSEM-BFM, whilst the Norwegian Trench is better represented by NEMO-ERSEM. GETM-ERSEM-BFM has more spatial variability than NEMO-ERSEM, despite having a lower model resolution. As with SST, there is notable inter-annual variability in the observations, with higher chlorophyll concentrations in 2011 than 2010. Both models captured this variability, although it is less evident in GETM-ERSEM-BFM, and over-pronounced in NEMO-ERSEM.

The models are similarly compared to the OC5 SPM products in Fig. 8. NEMO-ERSEM and GETM-ERSEM-BFM both under-estimate SPM in the central and northern North Sea, with NEMO-ERSEM also under-representing the plume of SPM off south-east England. Overall, the two models give very different results for SPM, and the reasons for and potential consequences of this are discussed in Sect. 5.

Maps of mean surface nitrate, phosphate and silicate for each model for August 2010 are shown in Fig. 9, alongside the

corresponding World Ocean Atlas climatology fields. Only 2010 is plotted because very similar patterns are seen in the models

for both years, and the climatologies do not include inter-annual variability. It should also be noted that the climatologies are of relatively coarse 1° resolution, so provide only a basic representation of the North Sea, but are the only source of data with full spatial coverage available for such a comparison. The climatologies have been used as boundary conditions by both models, so are not strictly independent, but values within the North Sea domain have not been used as input to the models.

For nitrate, the main limiting nutrient in the North Sea, GETM-ERSEM-BFM shows high coastal concentrations, and very low concentrations elsewhere. NEMO-ERSEM has a similar pattern, but with a much less extreme range of values. Likewise for phosphate and silicate, NEMO-ERSEM and GETM-ERSEM-BFM show differing distributions to each other, and match some of the climatological features but not others. Overlaid on the maps are in situ surface nutrient observations sampled on the 2010 IBTS cruise. These show near-depletion of nitrate and phosphate across most of the North Sea. The depletion of nitrate

is captured well by GETM-ERSEM-BFM, but is not seen to the same extent in either NEMO-ERSEM or the climatological World Ocean Atlas fields. The in situ observations also show greater depletion of phosphate than either of the models or the climatology, but the models are a better match for the silicate observations.

## 4.3 Model validation against PFT observations

This section presents validation of the models against the chlorophyll and PFT observations collected on the IBTS cruises.

Hereafter, Micro is used to refer to the fraction of total chlorophyll represented by the micro-phytoplankton size class, and similarly Nano and Pico. Model PFTs were aggregated as described in Sect. 3.4. The most northerly of the IBTS observations were located outside the GETM-ERSEM-BFM model domain; these have been excluded from the assessment to ensure the same observations were used to validate both models.

Maps of mean surface model PFTs for August 2010, as fractions of total chlorophyll, are plotted in Fig. 10. These show very

different distributions for NEMO-ERSEM and GETM-ERSEM-BFM, much more so than the difference in total chlorophyll might suggest. In particular, NEMO-ERSEM shows dominance by pico-phytoplankton in the southern North Sea, similar fractions of pico-phytoplankton and flagellates in the rest of the domain, and generally low concentrations of diatoms and dinoflagellates. In contrast, GETM-ERSEM-BFM shows dominance by diatoms in coastal regions, and by pico-phytoplankton in the centre of the domain, with generally lower fractions of the remaining PFTs. The two PFTs unique to GETM-ERSEM-

BFM, resuspended benthic diatoms and *Phaeocystis* colonies, only show notable concentrations in certain coastal areas. The reasons for the differences between the two models are discussed in Sect. 5.

The bias, root mean square error (RMSE) and correlation of modelled versus observed total chlorophyll concentration are shown in Table 3. GETM-ERSEM-BFM has a slight bias towards too high chlorophyll, whilst the bias for NEMO-ERSEM is near-zero. GETM-ERSEM-BFM chlorophyll values are typically higher than those from NEMO-ERSEM, but also show

a greater range. These features are consistent between the two years. However, whilst GETM-ERSEM-BFM has a similar correlation value for both years, the correlation for NEMO-ERSEM is much higher in 2010 than 2011. It should be noted though that these statistics are based on a relatively small number of points, so any conclusions drawn from this comparison are not guaranteed to be robust, particularly given the domain-scale spatial variability (see Sect. 4.2 and Fig. 7).

A comparison of phytoplankton community structure in the models and IBTS observations has been made by aggregating the model PFTs into the three observed PSCs, as described in Sect. 3.4. The bias, RMSE and correlation of the modelled versus observed total PSCs are shown in Table 3. Bias and RMSE are generally lower for GETM-ERSEM-BFM than for NEMO-ERSEM, particularly for Micro and Pico. NEMO-ERSEM has higher absolute correlations, but for Micro and Pico these tend to be negative, suggesting that NEMO-ERSEM is getting Nano approximately correct, but Micro and Pico are inversely distributed compared with the observations.

The distribution of relative PSC fractions with total chlorophyll is plotted for each data set in Fig. 11. Consistent with results from previous studies (e.g. Devred et al., 2011), as observed chlorophyll increases, Micro tends to increase, and Nano and Pico decrease. This pattern is also seen to some extent in GETM-ERSEM-BFM, but less so in NEMO-ERSEM (and only in 2010), although NEMO-ERSEM has a smaller range of chlorophyll concentrations. In the IBTS data there is a clear overall dominance of Micro. This is well reproduced by GETM-ERSEM-BFM, but the opposite is found in NEMO-ERSEM. The exception to this is a group of observations at low chlorophyll concentrations, most notably in 2011, in which Micro is least abundant, better matching typical NEMO-ERSEM results. These observations were all taken in the central North Sea, and this behaviour is discussed further in Sect. 5.

To explore the model behaviour further, and allow comparison with other works such as de Mora et al. (2016), histograms of the distribution of relative PSC fractions with total chlorophyll are plotted in Fig. 12, from each model grid point in the North Sea. These have used the mean model fields for August 2010 and August 2011. With this extended number of model points, a clear relationship is seen for GETM-ERSEM-BFM, with Micro increasing with total chlorophyll, and Pico decreasing. This matches the trend seen in the IBTS observations, as well as previous studies (e.g. Brewin et al., 2010; Devred et al., 2011). For NEMO-ERSEM, the range of chlorophyll concentrations remains small, making any relationship difficult to assess. When there are higher chlorophyll values a similar pattern of increasing Micro and decreasing Pico is seen, but there are too few points to draw a robust conclusion on the model relationship.

Three variables which always sum to one can be displayed in a single space, barycentric coordinates, using a ternary plot (e.g. Jupp et al., 2012). Phytoplankton community structure is plotted this way in Fig. 13. The observations form a distinct line in this space, from the centre of the plot to the corner representing dominance by Micro. At lower chlorophyll concentrations (not shown in Fig. 13, but consistent with Fig. 11) there are roughly equal fractions of Micro, Nano and Pico. As chlorophyll concentration increases, Nano and Pico decrease in roughly equal amounts, with Micro increasing accordingly. The fact that the observations form a line in this space shows Nano and Pico to change in roughly equal proportions when Micro changes with chlorophyll, which differs to some extent from other studies such as Brewin et al. (2010). GETM-ERSEM-BFM displays a similar pattern, with values in the same area of the plot as the observations, although with a less distinct relationship. NEMO-ERSEM values show a very different distribution however. There is some overlap with the observations in 2011, but otherwise a much less Micro-dominated regime is evident.

The ternary plot can also be used as a colour key to produce a map of phytoplankton community structure, as in Fig. 14. This plots the August mean community structure for each model and each year, overlaid with the IBTS point observations in circles. Plotting the community structure in such a fashion demonstrates that whilst GETM-ERSEM-BFM and NEMO-ERSEM give

very different results in terms of the magnitudes of the PSC fractions, there are nonetheless some broadly consistent features in terms of spatial patterns, which are also evident to some extent in the observations. For instance, both models (although NEMO-ERSEM less so in 2010) show a distinct split in community structure between the southern and northern North Sea, and around bathymetric features such as Dogger Bank and coastlines. Such a split can be clearly seen in the 2011 observations, which show

very little variation throughout the central North Sea, but is less clear in the 2010 observations. A difference between the years in the community structure in the central North Sea is also seen in NEMO-ERSEM, and to a lesser extent GETM-ERSEM-BFM, although the direction of change in the models is from Pico-dominated to Micro-dominated, the opposite of that in the observations. Although in most cases the community structure in the models does not match that of the observations, GETM-ERSEM-BFM is a very good match for the observations in the southern North Sea, an area particularly dominated by diatoms

in the model. Silicate in this region is near-depleted in GETM-ERSEM-BFM (see Fig. 9), but abundant in NEMO-ERSEM. In GETM-ERSEM-BFM, distinct blue patches can be seen off East Anglia, south Dorset and the German Bight, which are mostly areas where *Phaeocystis* colonies dominate in the model.

## 5   Summary and discussion

This study has presented a new set of in situ phytoplankton pigment observations for the North Sea, processed to give informa-

tion on phytoplankton community structure. Two physical-biogeochemical models, the biogeochemical components of which are different variants of ERSEM, were then validated against these and other observations. Both models were a good match for SST observations, and a reasonable match for chlorophyll observations, but gave contrasting results for SPM. Furthermore, the two models displayed very different phytoplankton community structures. GETM-ERSEM-BFM was able to reproduce many of the features of the observations, particularly in the southern North Sea, whereas NEMO-ERSEM was a poor match for the

observations, except at the lowest chlorophyll concentrations. Nonetheless, both models shared some similarities with each other and the observations in terms of spatial features and inter-annual variability.

The distribution of total phytoplankton biomass across the North Sea during summer of both years showed a high degree of consistency between three different observational methods: satellite remote sensing, high-frequency continuous measurement of chlorophyll fluorescence, and chl-a quantification at discrete stations. A similar set of spatial features can be observed in 2010

and 2011, which can be explained by the underlying hydrodynamics (van Leeuwen et al., 2015). The central, strongly-stratified region of the North Sea has very low nutrient concentrations and correspondingly low chl-a. In areas where vertical mixing, riverine input or horizontal advection bring nutrients into the upper water column, phytoplankton biomass is elevated. As well as observing the total quantity of phytoplankton, deriving the composition of size- and functional types is important for a better understanding of ecosystem function and energy flows to higher trophic levels (Chavez et al., 2011). Accessory pigments have

been widely used in biological oceanography to investigate community composition, but caution must be applied when interpreting results, and support from other methods should be used where possible (Schlüter et al., 2014). The original equations used by Hirata et al. (2008, 2011) to convert pigments to pico-, nano-, and micro-phytoplankton size classes underestimated the fraction of pico-phytoplankton compared to flow cytometric observations, and were modified by increasing the contribution of

chl-b and 19-hex to this class. Results in both years showed a consistent pattern of decreasing micro-phytoplankton abundance with distance from the coast, and with increasing latitude, and this is supported by previous pigment-based studies. Work in the German Bight has also shown a change from a coastal, diatom-dominated community to a more diverse, small-celled community further offshore (Brandsma et al., 2013; Wollschläger et al., 2015).

Both NEMO-ERSEM and GETM-ERSEM-BFM have shown the ability to reproduce the physics and broad-scale biogeo-chemistry of the North Sea. However, results are more varied when considering specific aspects such as the phytoplankton community structure in August. In some ways this is to be expected, as this study has used existing model versions which have not been previously validated against or tuned to such observations. Furthermore, August is a challenging month to model in the North Sea, as evidenced by the reduced SST skill for this month compared with the seasonal average. This is because

simulating the details of the stratification, nutrient concentrations and therefore phytoplankton concentrations is dependent on having successfully simulated processes in previous months, as well as the processes seen during August. This is more important than at other times of year, because, being at the height of summer, there are no strong temporal gradients driving the response of the system, as in spring and autumn. As a result, the internal biogeochemical dynamics can play out most freely (both in reality and in the model). Nonetheless, this kind of specific information is in increasing demand, and if results are to

be provided to users then they should be understood and validated.

NEMO-ERSEM and GETM-ERSEM-BFM gave very different representations of SPM concentrations, which impacts ecosystem functioning through light limitation (see also the light susceptibility parameters, Table 2). NEMO-ERSEM uses the two-size class SPM model described by Sykes and Barciela (2012). This was implemented by Sykes and Barciela (2012) in the POLCOMS (Proudman Oceanographic Laboratory Coastal Ocean Modelling System; Holt and James, 2001; Holt et

al., 2005) physical framework, in which it gave skilful results compared with observations. However, the model has not yet received the same degree of tuning and development since being implemented in NEMO-ERSEM, which may explain the consistent under-estimation of SPM found in this study. Furthermore, the high vertical resolution of NEMO-ERSEM means that the settling velocities must sometimes be artificially limited when used by the SPM model, in order to avoid breaking the Courant-Friedrichs-Lewy (CFL) condition (Courant et al., 1928), thus reducing resuspension. Changes to the settling parame-

ters would be expected to lead to improvements. GETM-ERSEM-BFM uses an alternative SPM model (van der Molen et al., 2017), which only has one size class but includes resuspension by waves as well as currents, and which was developed within the GETM-ERSEM-BFM framework. This generally matches spatial distributions of SPM better, but often has concentrations which are extremely high or low compared with satellite data, leading to a degradation in some error statistics.

The starkest contrast between the model results presented in this study is in the simulated phytoplankton community struc-

tures, which differ far more than might be expected given the corresponding total chlorophyll concentrations. GETM-ERSEM-BFM gave a wider range of combinations of biomass in the three size classes resolved by the observations than NEMO-ERSEM did. This was also reflected in more spatial variability and stronger spatial gradients, which resulted in a better match of the coastal to offshore change in phytoplankton community structure evident in the observations, in which diatoms are particularly important. The limited biomass in the two additional PFTs in GETM-ERSEM-BFM (benthic diatoms and *Phaeocystis* colonies,

Fig. 10) suggests that these were not the primary cause of this difference in response. Two mechanisms are likely to play im-

portant roles in causing the differences: 1) the higher coastal nutrient concentrations in combination with the different nutrient affinity settings in GETM-ERSEM-BFM, allowing diatoms to out-compete other types, in contrast with identical nutrient affinity settings in NEMO-ERSEM; 2) the coincidence of diatoms with areas of high SPM concentrations in GETM-ERSEM-BFM (mostly absent in NEMO-ERSEM) in combination with greater light susceptibility of diatoms (again contrasting with uniform values in NEMO-ERSEM), giving them competitive advantage. Similar, but more subtle effects will modulate the response of the other PFTs in GETM-ERSEM-BFM. Overall, the more uniform parameter settings of NEMO-ERSEM promote a more uniform response of the PFTs, as is evident in the results. Further work, with dedicated experiments performed in a configuration designed for such a comparison, could be carried out to further investigate these differences, and this is discussed below.

GETM-ERSEM-BFM provided a better match for the IBTS observations of phytoplankton community structure than NEMO-ERSEM did. The exception to this was in the low-chlorophyll waters of the central North Sea, the region of the domain with the weakest currents and largest residence times. Here the community structure of the observations more closely resembled that typical of NEMO-ERSEM. There are indications that recent versions of NEMO-ERSEM, applied to the global ocean rather than to the North-West European Shelf Seas, perform better at reproducing observed community structures (de Mora et al., 2016). Together with the results presented here, this suggests that NEMO-ERSEM may be more representative of an open ocean environment, whereas the settings in GETM-ERSEM-BFM are better suited to the complex coastal environment of the North Sea.

Whilst there were large contrasts in the corresponding ratios of PSCs between all three data sets, there was more agreement between the data sets about the spatial patterns of community structure. For instance, each had contrasting structures between the southern and central North Sea, and in coastal areas. Furthermore, inter-annual variability in the central North Sea was clearly evident in the observations, and also each of the models. This can be compared to differences in SST between the two years and the two models, suggesting a set of physical drivers which the models were able to capture. Across much of the North Sea the SST was cooler in 2011 than 2010, as seen in Fig. 5. Typical mixed layer depths were also deeper in the models in 2011 (not shown), likely in response to increased wind speeds. This deepening of the mixed layer cooled the SST and brought more nutrients to the surface, increasing production and chlorophyll, as seen in Fig. 7. In NEMO-ERSEM, which showed the most inter-annual variability in phytoplankton community structure, all PFTs increased in chlorophyll, but dinoflagellates increased the most in percentage terms, shifting the community structure more towards Micro-dominance. This could be because the changes in temperature and prey availability favoured smaller predators more than larger ones, and an increase in the ratio of nitrate to ammonium best suited larger phytoplankton. This implies that even if models are currently unable to accurately represent the exact community structures, they can still be used to assess the distribution of different habitats, and when and where these may change.

Due to differences in locations, timescales and datasets, a robust comparison with results from other PFT modelling studies in the literature cannot be made at this stage. Nonetheless, a consideration of the types of results obtained using different modelling approaches is of value. Lewis et al. (2006) and Lewis and Allen (2009) both validated a similar version of the PML-developed ERSEM, coupled with the POLCOMS hydrodynamic model, against in situ observations of PFTs in the North-west European Shelf. Lewis et al. (2006) validated against Continuous Plankton Recorder (CPR; Richardson et al., 2006)

data, and found that whilst the model reproduced the main seasonal features of plankton succession, diatoms consistently bloomed too early, and there were some spatial differences, especially in the North Sea. This would seem consistent with the low diatoms seen in NEMO-ERSEM in August in this study. Lewis and Allen (2009) validated against a station in the Western English Channel, and found a poor match for phytoplankton variables when assessment was performed in observation

space, as has been done here, but a better match when assessing more general seasonal variability. On a global open-ocean scale, Gregg and Casey (2007) validated the NASA Ocean Biogeochemical Model (NOBM) against both in situ and remotely sensed estimates of PFTs, with a focus on coccolithophores, and described their results as "mixed", noting contrasting results elsewhere in the literature. Hirata et al. (2013) validated the MEM-OU model against remote sensing estimates of large and small phytoplankton, and found good agreement at basin scales, but which reduced at smaller scales. De Mora et al. (2016)

compared PFT distributions with chlorophyll from the most recent ERSEM version of Butenschön et al. (2016) to those obtained using various remote sensing algorithms applied to the model data, and found the model displayed similar properties at global scales. These studies (not exhaustive) all used models with a relatively small fixed number of specifically-parameterised PFTs. An alternative approach is to initialise the model with a large number of randomly-parameterised PFTs, with the best-suited PFTs naturally dominating (Follows et al., 2007). In a global setting this approach has been found to successfully

reproduce large-scale patterns of phytoplankton diversity in terms of organism size (Follows et al., 2007) and number of co-existing species (Barton et al., 2010). The ability of such a model to reproduce the short-term variability of a complex coastal environment such as the North Sea would make for an interesting future inter-comparison. In general, results in the literature suggest some success of different approaches at reproducing large-scale patterns of phytoplankton community structure, but with more detailed skill yet to be properly demonstrated.

Careful thought needs to be given therefore to what products and information can be offered to users, which address user and policy requirements with a sufficient level of skill (Hyder et al., 2015). Continual model developments will be required. A comprehensive review of the challenges faced is given by Holt et al. (2014), and results from this current study should further inform future model development. This will be particularly relevant in the context of the UK Shelf Seas Biogeochemistry (SSB) research programme (http://www.uk-ssb.org), in which Cefas and the Met Office are both participants. One of the aims of the

SSB programme is to create a common version of ERSEM to be used by the UK research community, by combining features of the two versions of ERSEM used in this study. An initial combined version is described by Butenschön et al. (2016), and this will be further developed within the SSB programme. The version of Butenschön et al. (2016) provides a major update to that of Blackford et al. (2004), which forms the basis of the NEMO-ERSEM version used in this study, and initial applications (Butenschön et al., 2016; de Mora et al., 2016; Ciavatta et al., 2016) have shown different phytoplankton community structures

to that obtained in this work. It is clear from this study that the details of the model components and parameterisations can lead to very different results, and validation against a range of data, using a range of methods, is vital throughout the model development cycle.

The assessment presented in this study suggests that the biogeochemical model parameterisations are important in controlling the phytoplankton community structure. However, due to the many differences in physical modelling environment and

experimental configurations, it cannot be ruled out that differences in the physics are responsible for the differences in phyto-

plankton community structure, and previous studies have found these to be important. For instance, Sinha et al. (2010) coupled a single marine biogeochemical model with two different global physical ocean models, and found contrasting phytoplankton community structures due to differences in mixing. A number of further experiments could be performed to investigate the differences in PFT response between the models, and develop improvements. These can make use of the two ERSEM versions used in this study and the new SSB-ERSEM, along with NEMO, GETM, and GETM's 1D counterpart GOTM (General Ocean Turbulence Model). SSB-ERSEM is compatible with the Framework for Aquatic Biogeochemical Models (FABM; Brugge-man and Bolding, 2014), allowing it to be coupled with either GOTM, GETM or NEMO. Building on this framework could allow the different ERSEM versions to be run with identical hydrodynamics in 1D and 3D, and similarly allow individual ERSEM versions to be run with different hydrodynamics. This would help identify the differences in results introduced by dif-ferent components of the system. Within this experimental framework, controlled differences to physical and biogeochemical model parameters could be made to investigate further. These would allow very efficient testing. However, the current field observations (quasi-synoptic spatial distribution for one month of the year) do not allow a fully robust model comparison. An observational time series resolving the seasonal cycle at one or more locations would be needed for this excercise. Recent developments of algorithms to derive PFTs from remote sensing data (Nair et al., 2008; Hirata et al., 2013) could benefit all these potential strands of further work.

A further development will be the assimilation of biogeochemical data. Ocean colour data assimilation is being increasingly utilised by the reanalysis and forecasting community (Gehlen et al., 2015), and has already been successfully demonstrated for ERSEM (Ciavatta et al., 2011, 2014, 2016). A suitable ocean colour assimilation scheme for operational purposes is being developed as a collaboration between the Met Office and PML, to be implemented in the SSB ERSEM version and run operationally as part of CMEMS. This will also give the opportunity to take advantage of the advent of remote sensing PFT/PSC products, incorporating such data into the assimilation and routine validation.

Information on the marine environment can come from three sources: in situ observations, remote sensing data, and models. These three sources are inter-linked and all are vital; sufficient scientific understanding of the North Sea and other environments cannot be obtained if any of these three sources are removed. Models provide complete 3D spatial and temporal coverage, can be used to simulate a range of hypotheses, and are relatively inexpensive. However, as demonstrated in this study, observations are necessary for the validation and development of models, and model data cannot be relied upon in isolation. Remote sensing data provide considerably greater observational coverage than in situ measurements, but this coverage is still limited to the sea surface and cloud-free conditions, and empirical algorithms based on in situ data are used in the construction of remote sensing products. These satellite data must be comprehensively ground-truthed against in situ observations if they are to be used with confidence. Continuing in situ observations are therefore required to under-pin model and remote sensing data, as well as to provide unique insights into the marine environment. In turn, modelling studies can be used to help inform sampling strategies of future observing programmes, to help provide value for money without sacrificing accurate scientific understanding.

Finally, it remains to answer the question at the heart of this paper: "Can ERSEM-type models simulate phytoplankton com-munity structure?" The evidence from this study suggests that ERSEM-type models have the potential to accurately simulate phytoplankton community structure, but certain model formulations and parameterisations are required to do so, and these

two ERSEM versions do not reliably do so at this stage. Appropriate model development, informed by detailed validation studies, appears to be a major but achievable challenge, and will help facilitate the wider application of marine biogeochemical modelling to wide-ranging user and policy needs.

*Acknowledgements.* Sampling of the North Sea was funded by the European Union project PROTOOL (EU FP7, Grant No. 226880). The authors wish to thank Brian Harley and Sophie McCully and the officers and crew of RV *Cefas Endeavour* for their assistance during the IBTS surveys. David Ford and Rosa Barciela were funded through the Met Office Innovation Fund. David Ford, Rosa Barciela and Robert McEwan also received funding from the European Community's Seventh Framework Programme FP7/2007-2013 under grant agreement no. 283367 (MyOcean2), and from the Copernicus Marine Environment Monitoring Service. This paper is a contribution to the NERC-Defra funded Marine Ecosystems Research Programme (NERC award NE/L002981/1). Johan van der Molen and Kieran Hyder were funded through Cefas Seedcorn project DP235. The authors would like to thank the three anonymous reviewers and Dr Momme Butenschön for their comments in Biogeosciences Discussions.

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

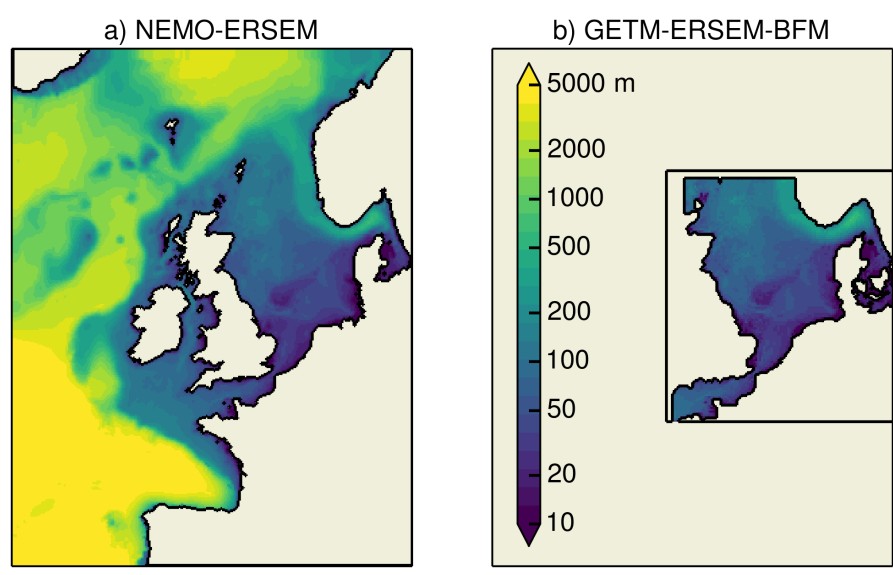

**Figure 1.** Maps of the model domain and bathymetry for a) NEMO-ERSEM and b) GETM-ERSEM-BFM.

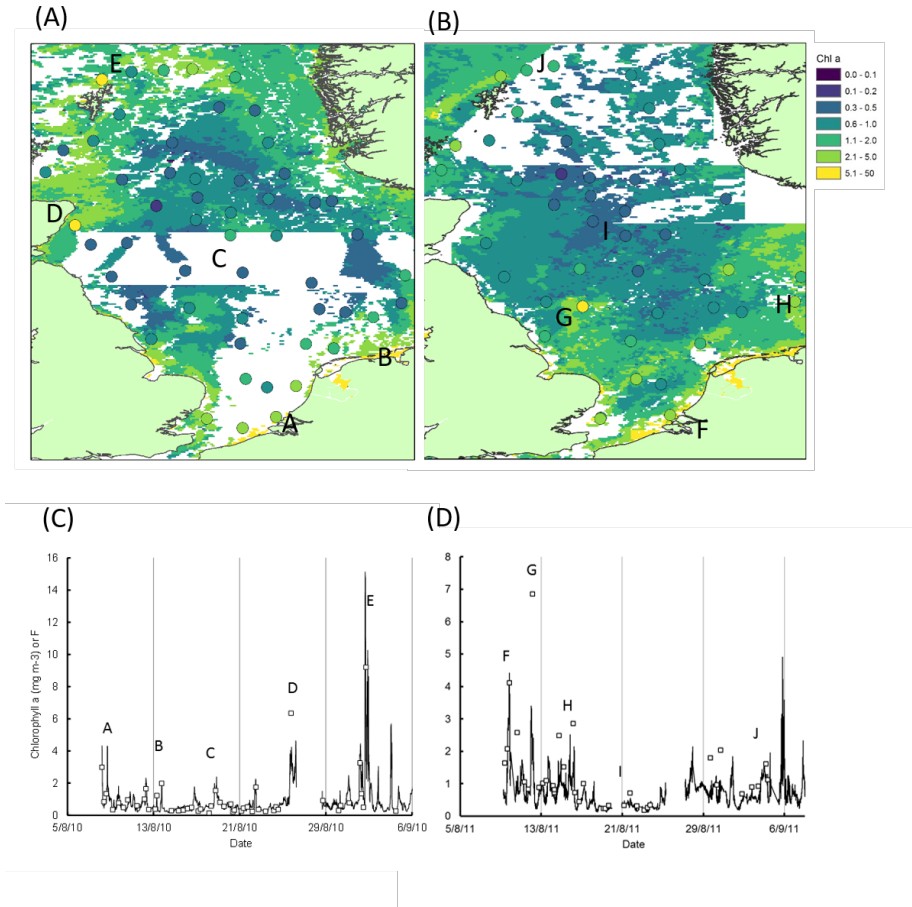

**Figure 2.** Maps of satellite-derived surface phytoplankton chlorophyll distribution during the summer IBTS cruises of a) 2010 and b) 2011, overlaid with the in situ observations in circles. White areas are where no satellite data were available. Time series of phytoplankton chlorophyll along the IBTS cruise track in c) 2010 and d) 2011 as assessed by continuous measurements of chlorophyll fluorescence (solid black line), and sampling of surface water for quantification of chl-a (open squares). Specific events along the tracks are referenced with a letter.

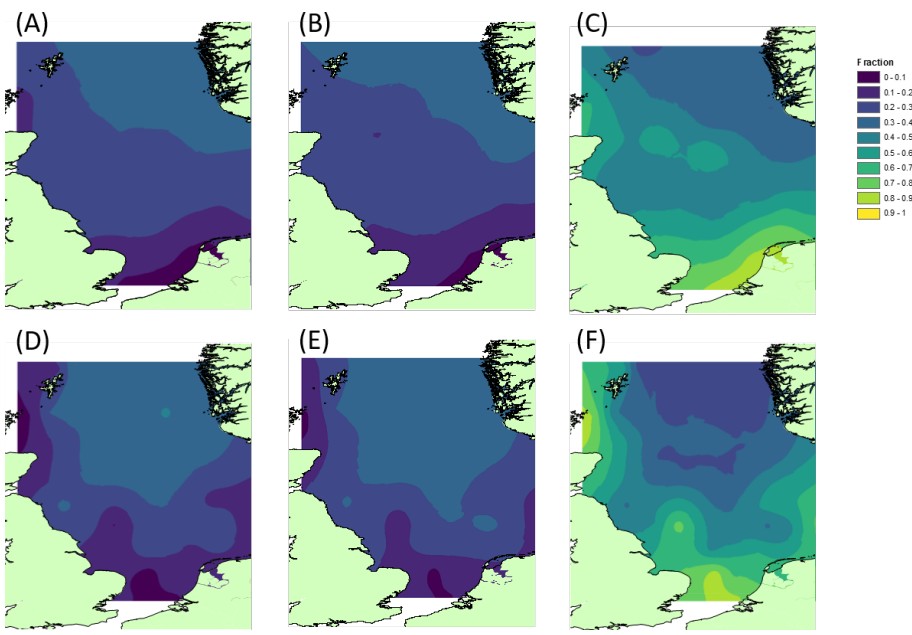

**Figure 3.** Maps of percentage surface PFT distribution during the summer IBTS cruises of 2010 (upper maps, a-c) and 2011 (lower maps, d-f) for pico-phytoplankton (a, d), nano-phytoplankton (b, e) and micro-phytoplankton (c, f).

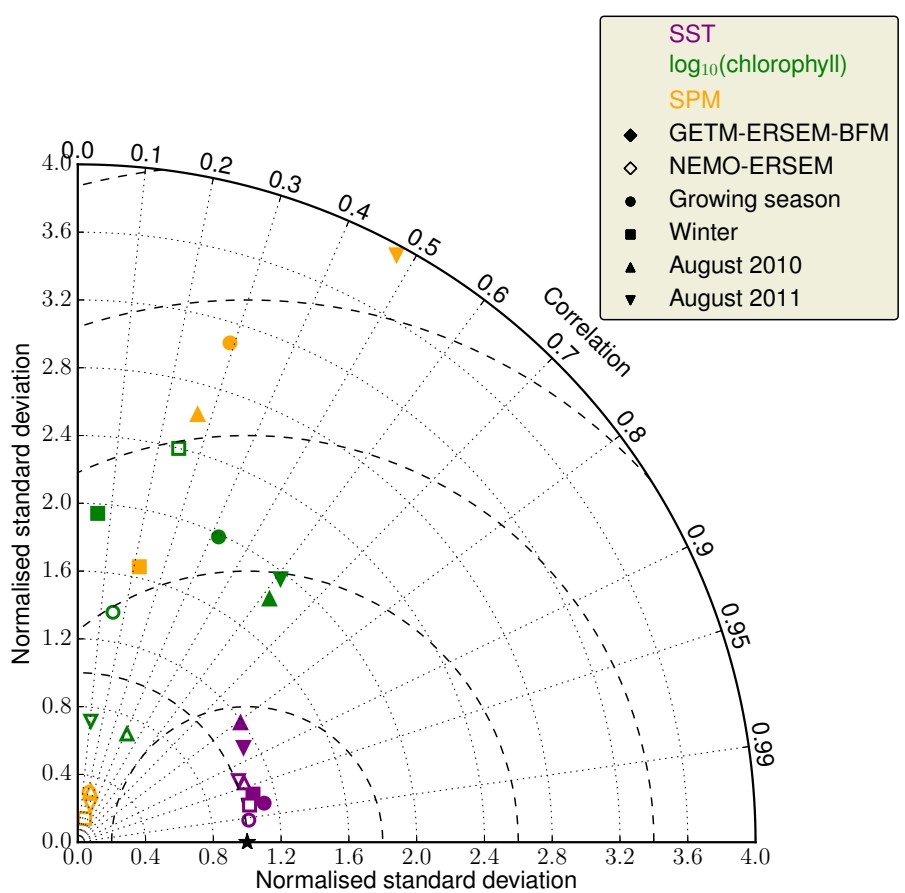

**Figure 4.** Taylor plot of SST (purple), $\log_{10}$(chlorophyll) (green), and SPM (orange) for the growing season (circles), winter (squares), August 2010 (upwards triangles), and August 2011 (downwards triangles). Filled symbols are GETM-ERSEM-BFM, unfilled symbols are NEMO-ERSEM. Validation has been performed in observation space, validating the models against OSTIA for SST, and OC5 ocean colour data for $\log_{10}$(chlorophyll) and SPM. A perfect model would plot at 1.0 on the x-axis, marked with a black star.

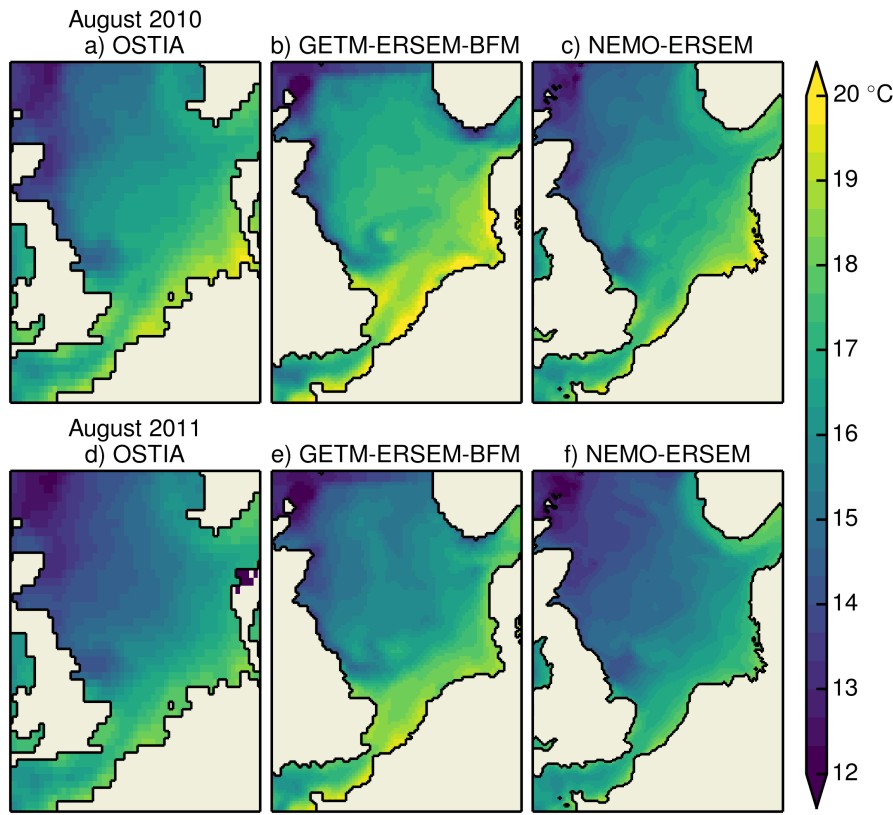

**Figure 5.** Maps of monthly mean SST for August 2010 (a-c) and August 2011 (d-f): observational data (a, d), GETM-ERSEM-BFM (b, e), and NEMO-ERSEM (c, f).

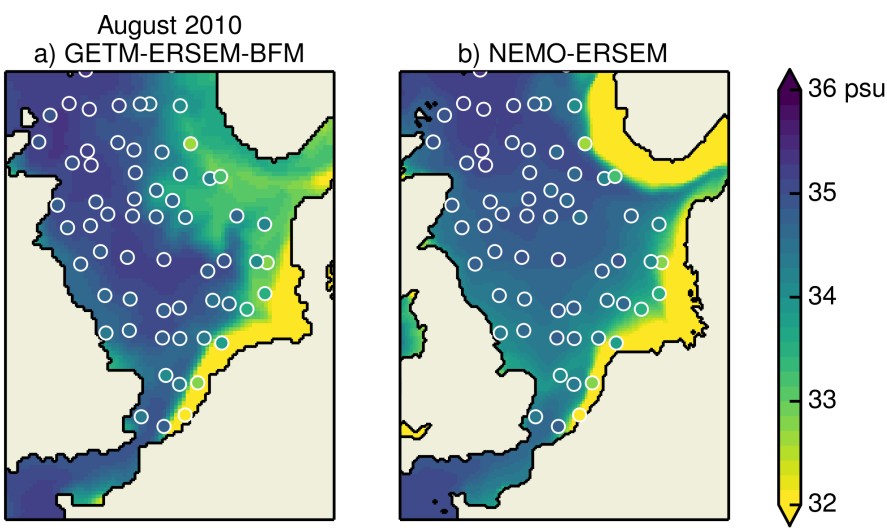

**Figure 6.** Maps of monthly mean SSS for August 2010 from a) GETM-ERSEM-BFM and b) NEMO-ERSEM. The in situ IBTS observations from August 2010 are overlaid in circles.

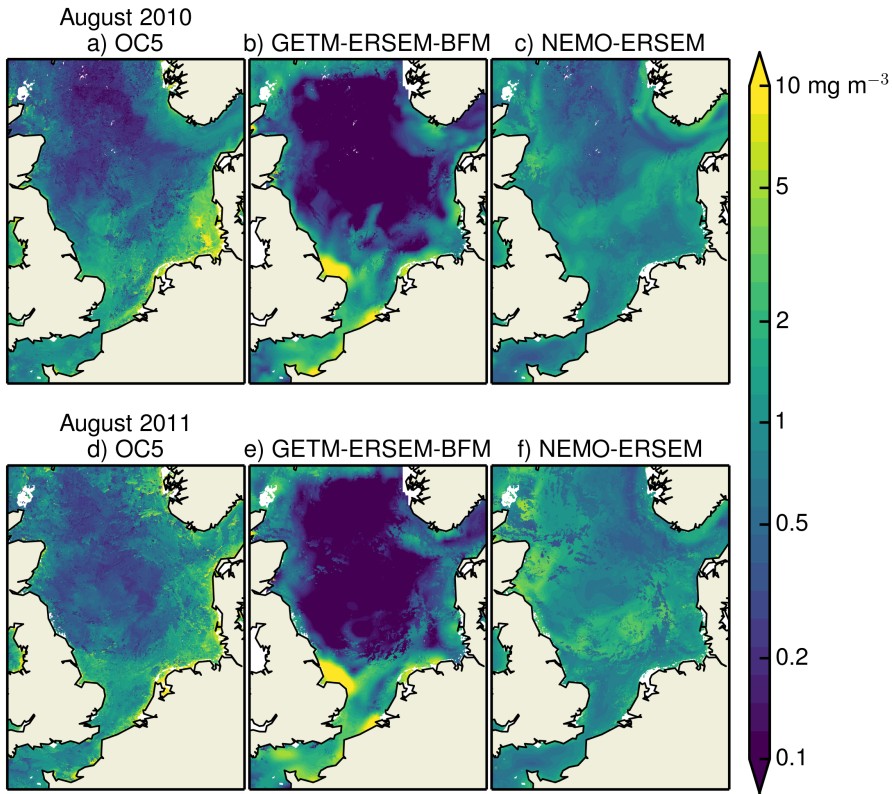

**Figure 7.** Composites of sea surface chlorophyll at ocean colour observation points for August 2010 (a-c) and August 2011 (d-f): satellite observations (a, d), GETM-ERSEM-BFM (b, e), and NEMO-ERSEM (c, f).

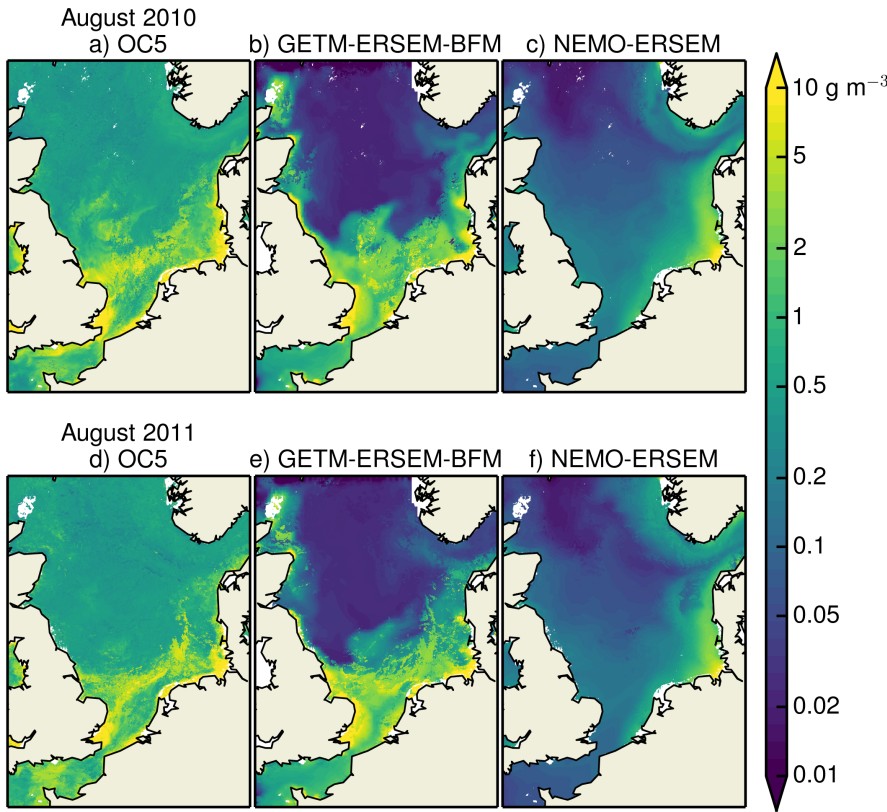

**Figure 8.** Composites of sea surface SPM at ocean colour observation points for August 2010 (a-c) and August 2011 (d-f): satellite observations (a, d), GETM-ERSEM-BFM (b, e), and NEMO-ERSEM (c, f).

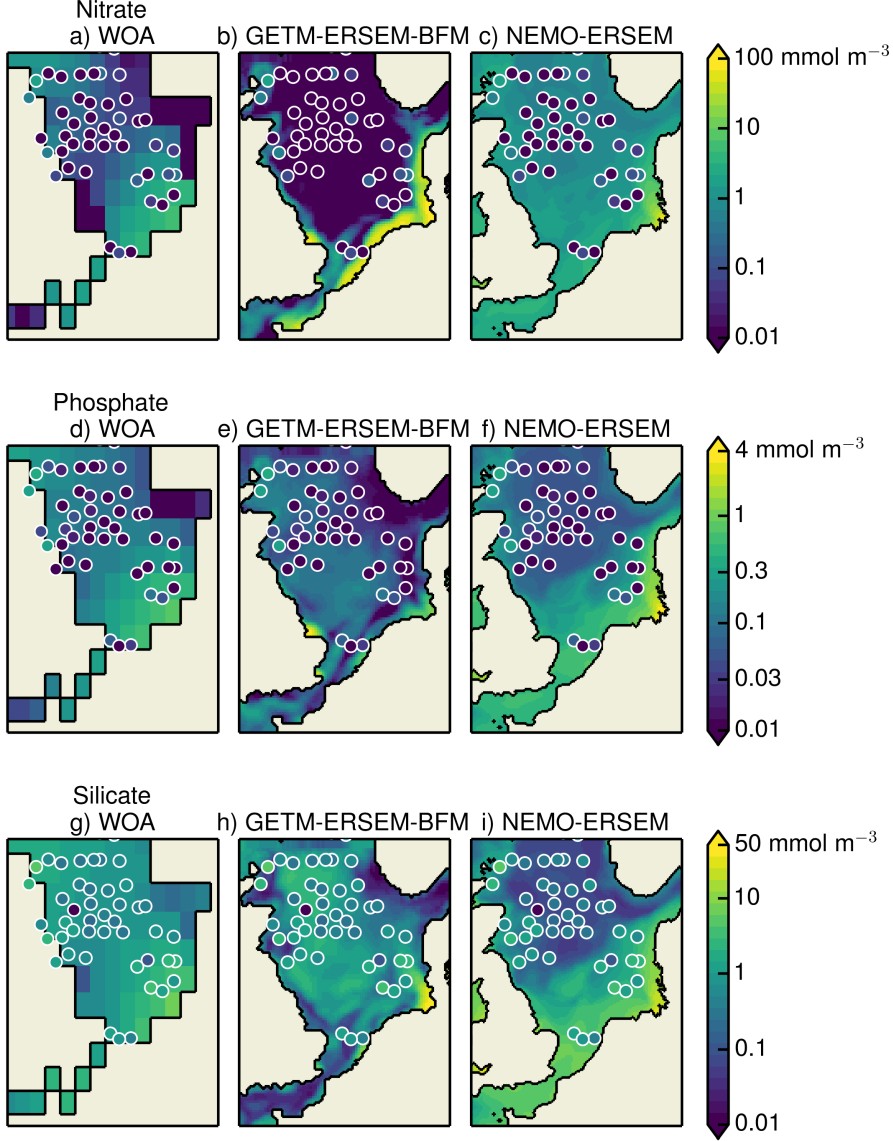

**Figure 9.** Maps of monthly mean sea surface nitrate (a-c), phosphate (d-f), and silicate (g-i) for August 2010: World Ocean Atlas climatology (a, d, g), GETM-ERSEM-BFM (b, e, h), and NEMO-ERSEM (c, f, i). The in situ IBTS surface observations from August 2010 are overlaid in circles.

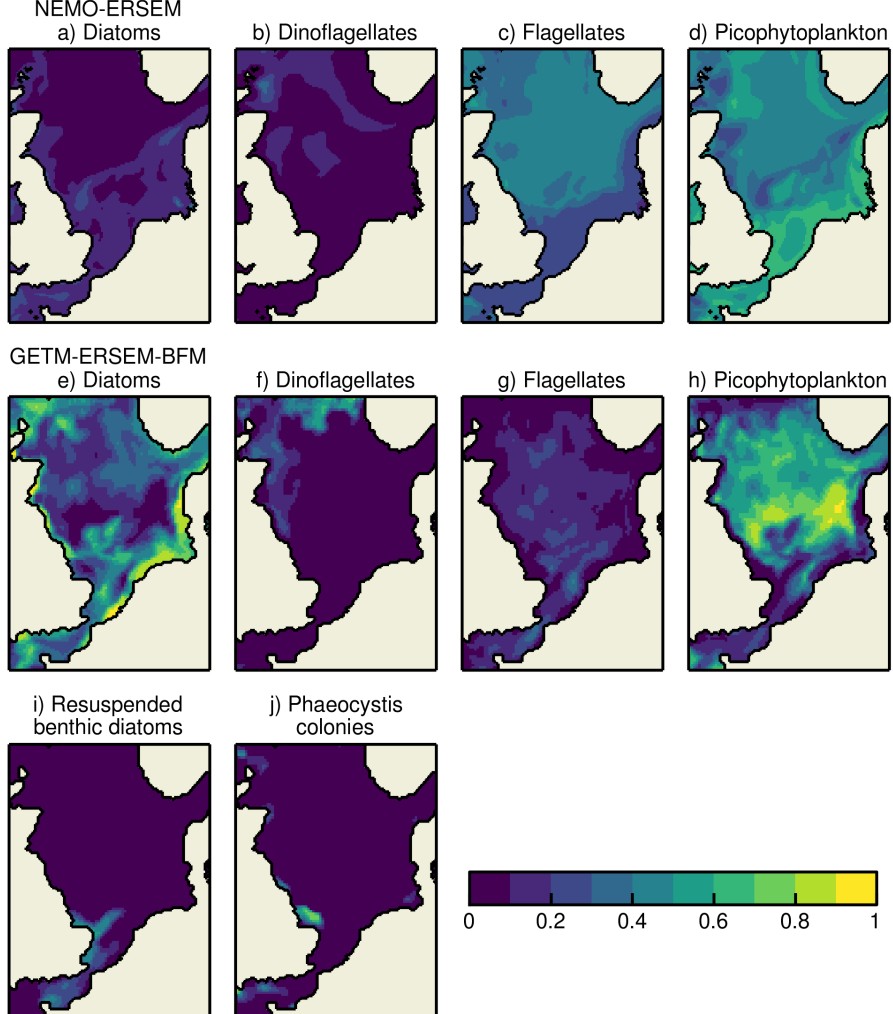

**Figure 10.** Maps of monthly mean sea surface PFT fractions for August 2010 from NEMO-ERSEM (a-d) and GETM-ERSEM-BFM (e-j). PFT fractions have been calculated as the proportion of the total sea surface chlorophyll concentration.

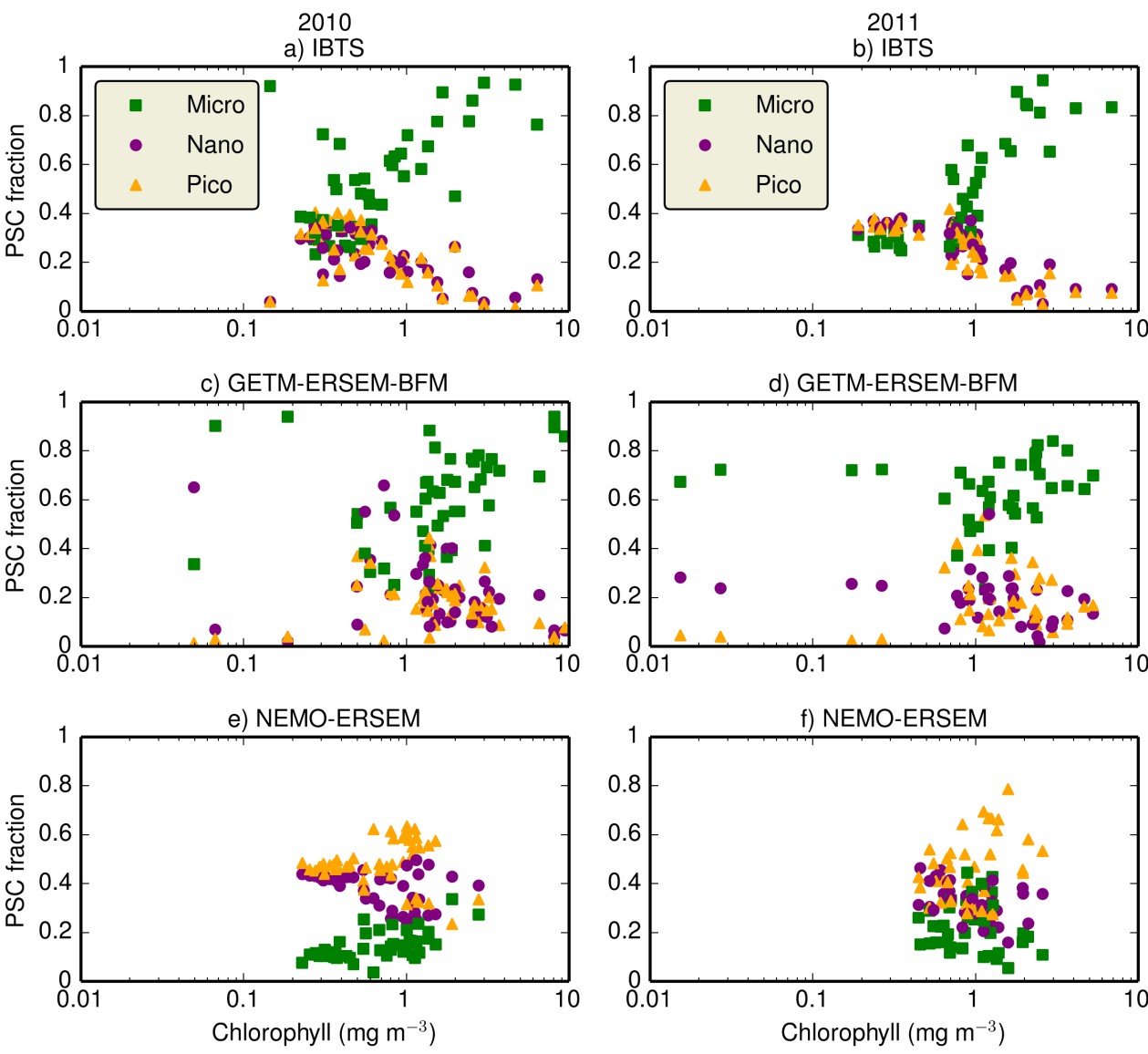

**Figure 11.** Phytoplankton size class (PSC) distribution as a function of chlorophyll concentration, plotted at each IBTS observation location in 2010 (a, c, e) and 2011 (b, d, f) from the IBTS observations (a, b), GETM-ERSEM-BFM (c, d), and NEMO-ERSEM (e, f).

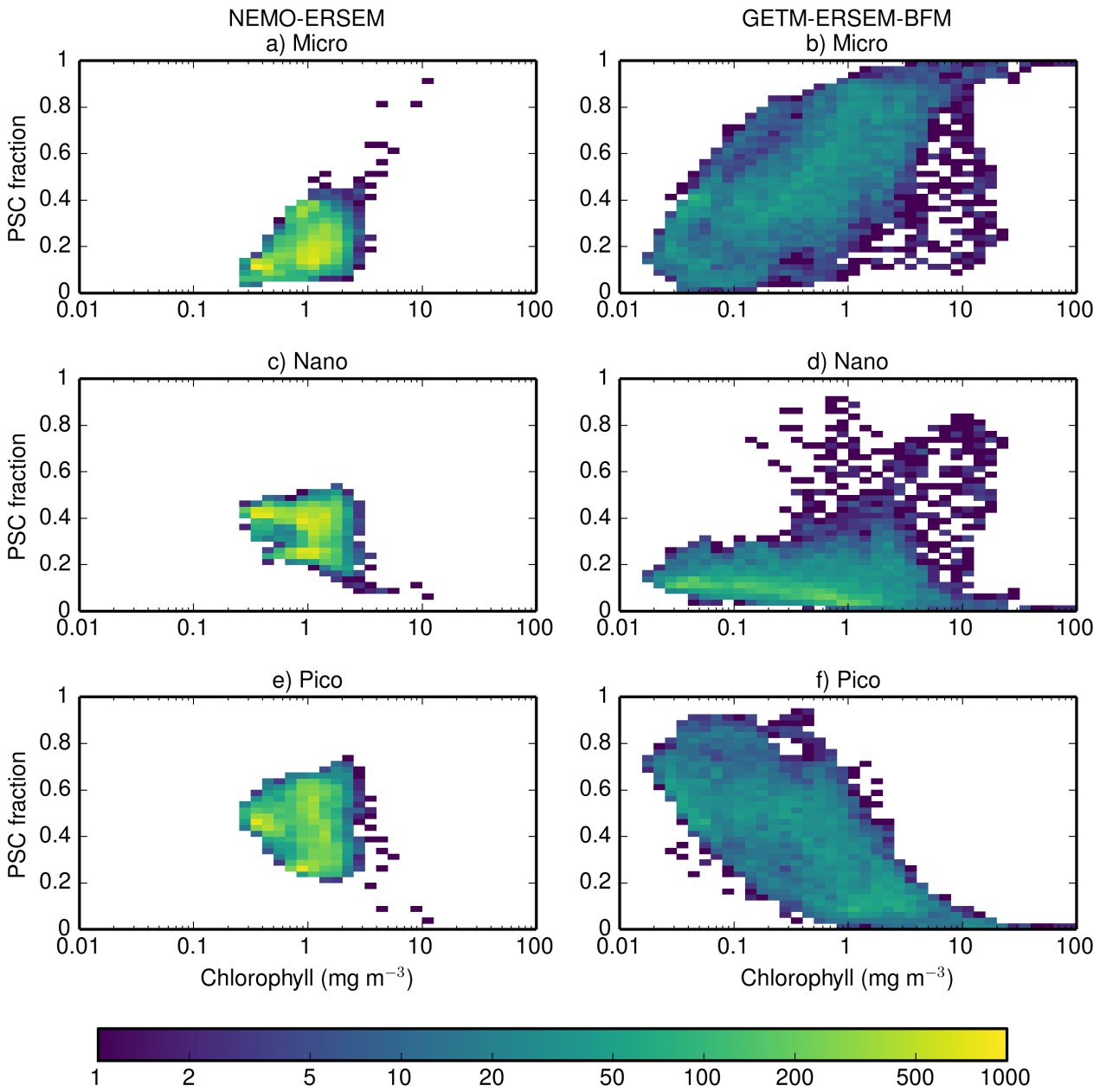

**Figure 12.** 2D histograms of the distribution of each phytoplankton size class (PSC) with chlorophyll, from NEMO-ERSEM (a, c, e) and GETM-ERSEM-BFM (b, d, f). The PSC fractions have been calculated at each surface model grid point in the monthly means for August 2010 and August 2011. Colours represent the number of occurences per bin.

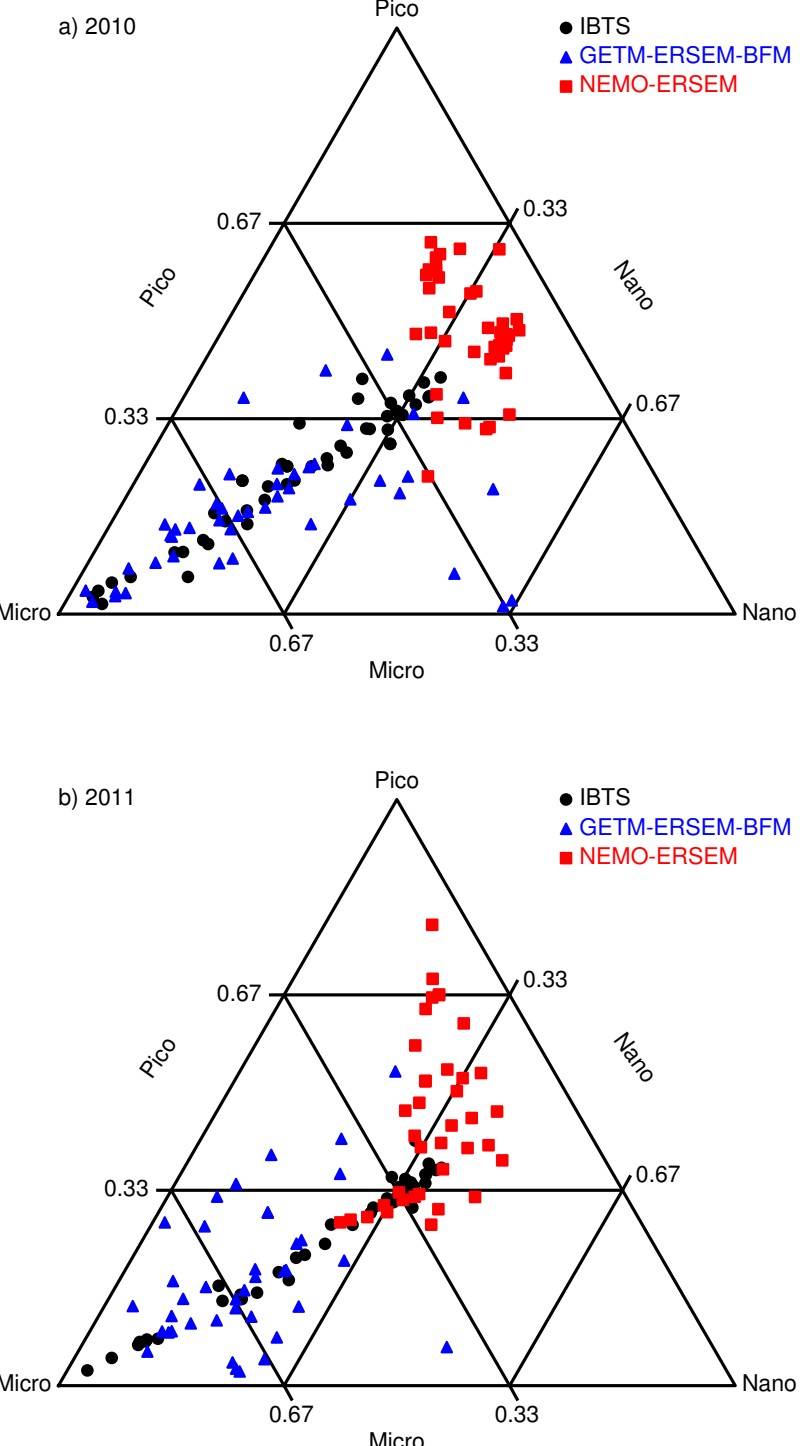

**Figure 13.** Ternary plots of phytoplankton community structure, showing the phytoplankton size class distribution at IBTS observation locations in a) 2010 and b) 2011, from the IBTS observations, GETM-ERSEM-BFM, and NEMO-ERSEM.

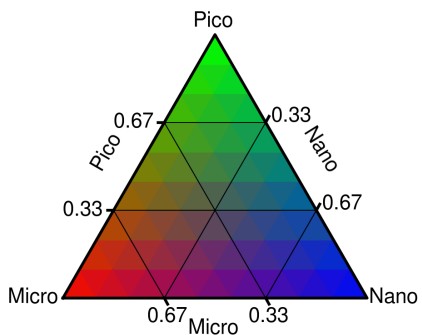

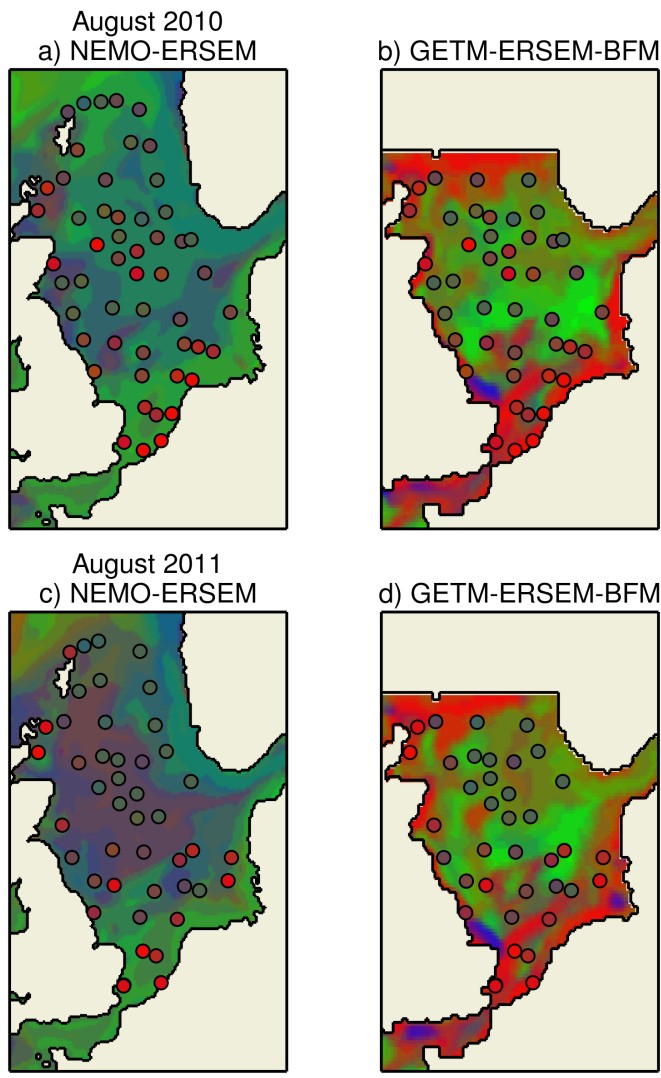

**Figure 14.** Maps of modelled surface phytoplankton community structure for August 2010 (a, b) and August 2011 (c, d) from NEMO-ERSEM (a, c) and GETM-ERSEM-BFM (b, d). The IBTS observations, sampled in August and early September of each year, are overlaid in circles.

**Table 1.** Summary of general-level model characteristics.

| | NEMO-ERSEM | GETM-ERSEM-BFM |
|---|---|---|
| Hydrodynamics | NEMO | GETM |
| Biogeochemistry | ERSEM | ERSEM-BFM |
| Domain | North-west European Shelf | North Sea |
| Horizontal resolution | ~7 km | ~10 km |
| Vertical resolution | 50 levels, terrain-following with constant 1m top box | 25 levels, terrain following General Vertical Coordinates |
| Tidal boundary | Elevation and currents from Met Office global model, Flather radiation condition | Elevations and currents from shelf model, Flather radiation condition |
| Temperature and salinity boundary | Met Office global model (GloSea5 reanalysis) | ECMWF global model |
| Nutrients boundary | World Ocean Atlas climatology | World Ocean Atlas climatology |
| Meteorological forcing | ECMWF ERA-Interim reanalysis: surface temperature, 2m wind, air pressure, heat fluxes, precipitation. 3-hourly | ECMWF ERA-40 and operational hindcast: surface temperature, 10m wind, air pressure, humidity, cloud cover. 6-hourly |
| Atmospheric nutrient deposition | Not included | Not included |
| River inputs | Freshwater flow: E-Hype; nutrients: climatology; sediments: daily climatology of satellite SPM at river points | Cefas data base, interpolated daily values of runoff and nutrients based on various observational sources |
| SPM concentrations | Modelled, two size classes, full transport, resuspension, aggregation and disaggregation | Modelled, 1 size class with concentration-dependent settling, full transport, resuspension by waves and currents |
| Nutrients | N, P, Si, C, O (Fe available but not used) | N, P, Si, C, O reduction equivalents |
| Pelagic autotrophic types | Diatoms, flagellates, dinoflagellates, picophytoplankton | Diatoms, flagellates, dinoflagellates, picophytoplankton, *Phaeocystis* colonies, resuspended benthic diatoms, pelagic nitrifiers |
| Zooplankton functional types | Mesozooplankton, microzooplankton, heterotrophic nanoflagellates | Filterfeeder larvae, mesozooplankton, omnivorous mesozooplankton, microzooplankton, heterotrophic nanoflagellates |
| Pelagic bacteria | Pelagic bacteria | Pelagic bacteria |
| Pelagic detritus | Labile dissolved organic matter, semi-labile dissolved organic matter, small particulate organic matter, medium particulate organic matter, large particulate organic matter | Labile organic carbon, TEP, particulate organic carbon (POC). Degradability of POC depends on nutrient:C quota. Vertical exchange of POC coupled to SPM transport |
| Type of benthic model | 3-layer model: oxic layer, denitrification layer, anoxic layer | 3-layer model: oxic layer, denitrification layer, anoxic layer |
| Seabed characterisation | Distribution of the two modelled SPM size classes, dependent on model dynamics | Porosity interpolated from North Sea Benthos Survey grain size data |
| Benthic autotrophic types | Not included | Benthic diatoms, benthic nitrifying bacteria, nitrifying archaea |
| Benthic macrofauna | Deposit feeders, suspension feeders, meiobenthos | Epibenthos, Deposit Feeders, Filter Feeders, Meiobenthos, Benthic predators |
| Benthic bacteria | Aerobic bacteria, anaerobic bacteria | Aerobic bacteria, anaerobic bacteria |
| Benthic detritus | Dissolved organic matter, particulate organic matter, buried organic matter | Labile organic carbon, particulate organic carbon |
| $CO_2$ method | Available but not used | Benthic and pelagic $CO_2$, pH, alkalinity |
| Pelagic nutrient regeneration | Nitrification depends on dynamics of nitrifying bacteria | Nitrification depends on dynamics of nitrifying archaea and bacteria |
| Benthic nutrient regeneration | Modelling of fluxes based on estimation of nutrient gradients on basis of processes and concentrations in the 3 benthic layers | Modelling of fluxes based on estimation of nutrient gradients on basis of processes and concentrations in the 3 benthic layers for phosphate, ammonium, nitrate, reduction equivalents, silicate, DIC, alkalinity. Dynamic determination of nitrification rate from benthic nitrifier model |
| Spinup period | Previous hindcast of Edwards et al. (2012), run for 2007 from previously spun-up fields | 1991-2001 |
| Production run | 2003-2012 (also run for 1983-1989 and 1989-2003, but sections not continuous) | 2002-2011 |
| Data assimilation | Satellite and in situ SST; 3D-Var | Not included |
| References | Blackford et al. (2004); Edwards et al. (2012) | Baretta et al. (1995); Ruardij and van Raaphorst (1995); van der Molen et al. (2016) |

**Table 2.** Parameters of the four coinciding phytoplankton functional types related to inter-species competition.

| | NEMO-ERSEM | | | | | GETM-ERSEM-BFM | | | | |
| --- | --- | --- | --- | --- | --- | --- | --- | --- | --- | --- |
| | Method/ parameter | Diatoms (P1) | Flagellates (P2) | Pico-phytoplankton (P3) | Dinoflagellates (P4) | Method/ parameter | Diatoms (P1) | Flagellates (P2) | Pico-phytoplankton (P3) | Dinoflagellates (P4) |
| Doubling temperature | $Q_{10}$ | 2 | 2 | 2 | 2 | $Q_{10}$ | 2 | 2 | 2 | 2 |
| Maximum productivity at 10 $^\circ$C | sum | 2.5 | 2.7 | 3.3 | 1.5 | sum | 3.0 | 4.875 | 5.6 | 1.75 |
| Respiration rate at 10 $^\circ$C | srs | 0.05 | 0.05 | 0.05 | 0.05 | srs | 0.125 | 0.1 | 0.1 | 0.125 |
| Fraction of PP excreted as PLOC/PDET | pu_ae | 0.05 | 0.2 | 0.2 | 0.05 | pu_ae | 0.05 | 0.1 | 0.1 | 0.05 |
| Activity respiration rate | pu_ra | 0.1 | 0.25 | 0.25 | 0.25 | pu_ra | 0.1 | 0.1 | 0.2 | 0.1 |
| Half-value of $SiO_4$-limitation | chP1sX | 0.3 | - | - | - | P_chPs | 0.3 | - | - | - |
| Minimum quota N | qnlP1cX ... qnlP4cX | 0.00687 | 0.00687 | 0.00687 | 0.00687 | p_qnlc | 0.00687 | 0.00687 | 0.00687 | 0.00687 |
| Minimum quota P | qplP1cX ... qplP4cX | 0.0004288 | 0.0004288 | 0.0004288 | 0.0004288 | p_qplc | 0.0003931 | 0.0003931 | 0.0003931 | 0.0003931 |
| Minimum quota Si | - | - | - | - | - | - | 0.09 | - | - | - |
| Multiplication factor for critical N:C ratio | xpcP1nX ... xpcP4nX | 1 | 1 | 1 | 1 | Not included | - | - | - | - |
| Multiplication factor for critical P:C ratio | xpcP1pX ... xpcP4pX | 1 | 1 | 1 | 1 | Not included | - | - | - | - |
| Multiplication factor for maximum quotum nitrate uptake | xqnP1X ... xqnP4X | 2 | 2 | 2 | 2 | p_xqn | 2 | 2 | 2 | 2 |
| Multiplication factor for maximum quotum phosphate uptake | xqpP1X ... xqpP4X | 2 | 2 | 2 | 2 | p_xqp | 2 | 2 | 2 | 2 |
| Multiplication factor for maximum quotum silicate uptake | - | - | - | - | - | p_xqs | 1.5 | - | - | - |
| Affinity for N(3) [nitrate] | quP1n3X ... quP4n3X | 0.0025 | 0.0025 | 0.0025 | 0.0025 | p_qun | 0.15 | 0.215 | 1.29 | 0.0084 |
| Affinity for N(4) [ammonium] | quP1n4X ... quP4n4X | 0.01 | 0.01 | 0.02 | 0.01 | Grouped with N(3) | - | - | - | - |
| Affinity for P | qurP1pX ... qurP4pX | 0.0025 | 0.0025 | 0.0025 | 0.0025 | p_qup | 0.15 | 0.215 | 1.29 | 0.0084 |
| Affinity for Si | qsP1cx | 0.03 | - | - | - | p_qus | 0.1 | 0 | 0 | 0 |
| Nutrient stress threshold for sinking | esNIP1X ... esNIP4X | 0.7 | 0.75 | 0.75 | 0.75 | Different method: based on TEP production | - | - | - | - |
| Sinking by formation of macro-aggregates | - | - | - | - | - | - | Threshold process: in presence of sufficient TEP and diatoms | Sticking to macro-aggregates | Sticking to macro-aggregates | Sticking to macro-aggregates |
| Lysis rate | sdoP1X ... sdoP4X | 0.05 | 0.05 | 0.05 | 0.05 + 0.2 | Max. by nutrient stress, p_sdmo | No lysis | 0.025 | 0.15 | 0.001 + 0.0 |
| Stress excretion | - | - | - | - | - | - | Extraction of TEP (carbo-hydrates) | - | - | - |
| Light susceptibility | P-I curve, alpha | 2.98 | 2.98 | 2.98 | 2.98 | P-I curve, Ke | 35 | 70 | 124 | 116 |
| Maximum Chl to C ratio | phimP1X ... phimP4X | 0.035 | 0.035 | 0.035 | 0.035 | p_qchlc | 0.02 | 0.035 | 0.035 | 0.035 |
| Minimum Chl to C ratio | phiP1HX ... phiP4HX | 0.025 | 0.025 | 0.025 | 0.025 | p_qlPlc | 0.0025 | 0.0035 | 0.0025 | 0.00225 |
| Background phytoplankton sinking rate | m day$^{-1}$ | | | | | m day$^{-1}$ | 0 | 0 | 0 | -5 |
| Maximum phytoplankton sinking rate | | | | | | m day$^{-1}$ | 90 | 90 | 90 | 90 |

**Table 3.** Statistical comparison of $\log_{10}$(chlorophyll) and phytoplankton size classes against IBTS observations.

| Year | Model | Variable | Bias | RMSE | Correlation | No. observations |
|---|---|---|---|---|---|---|
| 2010 | GETM-ERSEM-BFM | $\log_{10}$(chlorophyll) ($\log_{10}$(mg m$^{-3}$)) | 0.369 | 0.597 | 0.334 | 46 |
| | | Micro (fraction) | 0.069 | 0.285 | 0.010 | |
| | | Nano (fraction) | -0.006 | 0.170 | 0.116 | |
| | | Pico (fraction) | -0.062 | 0.182 | -0.265 | |
| | NEMO-ERSEM | $\log_{10}$(chlorophyll) ($\log_{10}$(mg m$^{-3}$)) | 0.017 | 0.339 | 0.446 | |
| | | Micro (fraction) | -0.385 | 0.434 | 0.160 | |
| | | Nano (fraction) | 0.146 | 0.165 | 0.604 | |
| | | Pico (fraction) | 0.240 | 0.292 | -0.369 | |
| 2011 | GETM-ERSEM-BFM | $\log_{10}$(chlorophyll) ($\log_{10}$(mg m$^{-3}$)) | 0.178 | 0.543 | 0.343 | 39 |
| | | Micro (fraction) | 0.138 | 0.261 | 0.265 | |
| | | Nano (fraction) | -0.076 | 0.134 | 0.387 | |
| | | Pico (fraction) | -0.062 | 0.178 | -0.061 | |
| | NEMO-ERSEM | $\log_{10}$(chlorophyll) ($\log_{10}$(mg m$^{-3}$)) | 0.067 | 0.389 | 0.157 | |
| | | Micro (fraction) | -0.270 | 0.382 | -0.416 | |
| | | Nano (fraction) | 0.063 | 0.121 | 0.359 | |
| | | Pico (fraction) | 0.207 | 0.299 | -0.530 | |
| 2010+2011 | GETM-ERSEM-BFM | $\log_{10}$(chlorophyll) ($\log_{10}$(mg m$^{-3}$)) | 0.282 | 0.573 | 0.320 | 85 |
| | | Micro (fraction) | 0.101 | 0.274 | 0.097 | |
| | | Nano (fraction) | -0.039 | 0.154 | 0.178 | |
| | | Pico (fraction) | -0.062 | 0.180 | -0.163 | |
| | NEMO-ERSEM | $\log_{10}$(chlorophyll) ($\log_{10}$(mg m$^{-3}$)) | 0.040 | 0.363 | 0.358 | |
| | | Micro (fraction) | -0.332 | 0.411 | -0.210 | |
| | | Nano (fraction) | 0.108 | 0.146 | 0.401 | |
| | | Pico (fraction) | 0.225 | 0.295 | -0.452 | |