# Peer review of "Observing and modelling phytoplankton community structure in the North Sea"

_Biogeosciences, 2016_

## Referee Comment (RC1) · Anonymous Referee #1 · 24 Aug 2016

The subject of the paper proposed by Ford et al. is of great interest for the marine ecologists but also for European managers, because very few distributed information is available about phytoplankton biodiversity and its changes caused by anthropogenic nutrient enrichment and warming of continental shelf waters. Contributing to fill this gap through 3D biogeochemical models requires new field measurements of phytoplanktonic functional types to validate the corresponding model state variables, as presented in this paper. Many of the authors of that paper are well-known for their previous important scientific contribution to ecological modelling of the North-West European Shelf.

The text is well written and the figures rather clear; the maps are small, but generally

usable. I suggest using more contrasted "pole" colors in the triangular representation used in figure 11.

As far as the models are concerned, a simple comparison between two versions of 3D-distributed ERSEM model is interesting, but rather difficult to be analysed, because both the physics and the biogeochemistry are different. As a caricatural example, the comparison between the assimilated temperature of NEMO-ERSEM and the free-running temperature of GETM-ERSEM-BFM (figure 4) seems to me inapplicable. Secondly, when strong differences appear, the authors should try to explain more what process (physical or biogeochemical) can be thought to be possibly the main driver.

The Taylor's diagram (Figure 3) shows (as in many models) that chlorophyll and SPM are not so realistically simulated, especially chlorophyll for the GETM-ERSEM-BFM version using the more complex biogeochemistry and sediment modelling, and SPM for the NEMO-ERSEM simpler version. The discrepancy in August chlorophyll is clearly visible on Figure 5: whereas GETM-ERSEM-BFM produces too much chlorophyll in the English Channel and the southern North Sea (especially this bloom in front of the Humber estuary) along with a too poor biomass in the central North Sea, NEMO-ERSEM has too smooth results and underestimates the chlorophyll on the coastal shallow German and Danish zones. For SPM, Figure 6 shows a good adequation for GETM-ERSEM-BFM, but a strong underestimation in the shallow waters for NEMO-ERSEM. So, I disagree a little bit with the authors when they say that "both models were still able to capture the main observed features throughout the domain." For nutrients, the too coarse maps of WOA climatologies don't allow a precise validation; but the authors could try to link the different behaviours of the two versions to their deposition and remineralisation processes : in the central North Sea, GETM-ERSEM-BFM seems to trap too much the detrital forms of N, P and Si. As far as biodiversity is concerned, GETM-ERSEM-BFM clearly gives a more realistic decomposition in phytoplanktonic functional types, while the weakness of diatoms in coastal nutrient-enriched zones seems to be strongly unrealistic in the case of NEMO-ERSEM, even in August.

[Figure]

The discussion could be highly improved by replacing some general considerations by a comparison with results obtained in PFTs simulation obtained by other scientists, especially those using the DARWIN model (Follows et al., 2007)

So, my general impression is that the subject of this paper is a good and actual one, but that the results of the two models used don't support really the optimistic answer given at the end of the paper by the authors: for me the ERSEM model has to be better calibrated before being considered as able to simulate biodiversity. Nevertheless, I think that this paper is acceptable as a milestone one on the long way to biodiversity modelling, but only if the authors add a bibliography-based discussion and lower their optimistic evaluation of their present versions of the model.

Literature cited:

Follows MJ, Dutkiewicz S, Grant S, Chisholm SW. 2007. Emergent biogeography of microbial communities in a model ocean. Science 315:1843-46

---

## Referee Comment (RC2) · Anonymous Referee #2 · 24 Aug 2016

General Comments:

The manuscript investigates the phytoplankton community structure and distribution in the North Sea (NS), employing two versions of ERSEM and a set of observations including remote-sensing and in-situ data. The study demonstrates a good match between different measurement techniques and narrates an extensive performance assessment of 2 model versions, and concludes that one of the ERSEM versions - the one which is used more for coastal applications, show higher skill then the other, which is usually used for a larger domain. Manuscript is clearly written and easy to follow. The study exhibits a high degree of technical sophistication: both versions of ERSEM provide detailed descriptions of numerous processes driving the highly complex ecosystem under investigation, and the observation dataset, especially regarding the HPLC-derived PFT's provide an excellent base for model evaluation. However, despite these ample resources and recognized expertise of authors, the manuscript fails to deliver a comparably significant outcome or a crisp message in its current form.

Specific Comments:

1) The title is not representative:

a) The title suggests this work is about biodiversity, but actually it's not. Although the 'relative abundance' of PFT's (Fig. 2,10 and 11) is an index relevant for biodiversity, the results are not at all discussed or framed in the context of biodiversity (in fact only the very last short paragraph of the manuscript refer to biodiversity again after the introduction, and replacing 'biodiversity' with 'phytoplankton community structure' would probably improve this paragraph already).

b) Again, at the first glance 'ERSEM-type models' looks like more general than it is, whereas the manuscript is literally about 2 ERSEM versions, which could be more clearly indicated in the title. Accordingly, I invite co-authors to consider a more representative title (as changing the content according to the current title would basically result in a new manuscript)

2) Validation of PFT estimates is not a new endeavor:

A recurring notion in the manuscript (abstract, introduction, summary and discussion) is that 'model estimates of PFTs are not validated', which is misleading. Quite a few attempts have been made previously to assess model performance with respect to reproducing the spatial distributions of PFTs, such as those making use of the data from continuous plankton recorder surveys, monitoring data obtained at stations located at different sites and estimates from satellite images, which should be acknowledged (examples are too many that pointing to specific studies would be unfair to others).

3) 'Observations' section is incomplete:

In this section, only the procedures relating to the HPLC data are provided. Other data sources, like the 'in-situ quantification of chl-a', OSTIA-SST, OC5-chlorophyll/SPM and WOA data should be introduced also in this section, and not elsewhere (e.g., as currently done in sections 4.1 and 4.2, see also the 'Technical Corrections')

4) 'Models' section is inadequate:

a) 3.1 GETM-ERSEM-BFM: each item in the list of additional processes to 'make [ERSEM as presented by Vichi et al. (2007)] suitable for temperate shelf seas' need a reference to previous literature if possible (like in the list items iv and v), or if not, an adequate description.

b) 3.2 NEMO-ERSEM: page(P)6,line(L)15-16: the employed version of NEMO appears to be not described previously (although a full description is foreseen to be provided by O'Dea at al., in prep), so here at least some hints should be provided

c) 3.3 Comparison: a figure showing the complete domains of the two models would be helpful

5) Fig.11 is difficult to perceive:

although I like the idea of overlaying measurements on the contour-plots using a ternary color map, I found it difficult to distinguish the resulting colors. Although I admit distinguishing colors is not my strength, I wonder whether using a more saturated and bright color scheme would help (eg., as in Fig.4 of Arteaga et al. 2014). The authors may also want to consider plotting the percentage histogram of each group across total chlorophyll as in Fig. 3 of De Mora et al. (2016), which does not rely at all on the color-perception abilities of the reader.

6) How to reproduce the spatial distribution of PFT's? :

as mentioned above, the results suggest that the ERSEM version previously tuned more for coastal applications (GETM-ERSEM-BFM) performs better than the one used for simulating the margin of the Atlantic Ocean (NEMO-ERSEM). But considering that

the performance difference wrt the PFTs basically boils down to the reproduction of the diatom blooms at the coast by GETM-ERSEM-BFM and not by NEMO-ERSEM, the obvious follow-up question is why the latter cannot reproduce the Diatom blooms at the coast, which I believe should be straightforward to figure out. It is hinted in section 3.3 and later in P15,L1-7 that the parameterization of the PFT's is likely to be the reason, but, it should be straightforward to test whether this explanation really holds. And I believe this would be a rather useful outcome of this study with a potentially wider relevance: can an open-ocean model transferred to a coastal system just by a reparameterization of the PFTs?

7) What drives inter-annual differences?:

Figures 4-11 suggest considerable differences between 2010 August and 2011 with respect to physical, chemical and biological characteristics. Although these differences are acknowledged throughout the manuscript, not much is done to elaborate the mechanisms driving these differences, apart from hinting 'a set of physical drivers' to be responsible for the difference (P15, L21). Again, I feel like more could be done with the available tools and data.

Technical Corrections:

- P6, L34: sediments=? SPM?

- P7, L3: nutrients =? of all forms, or dissolved inorganic form only?

- P7, L28,L30: first occurrence of the 'PML-ERSEM'. Although what this means should be clear to the careful reader, but better stick with the previous definitions.

- P9, L11: first occurrence of MODIS (also consider my specific comment (SC)-3)

- P9, L26-27: 'Picophytoplankton...' reference needed, and consider SC-3

- P10, L9-14: consider SC-3

- P10, L33-34: 'The largest PFT...' reference needed and consider SC-3

[Figure]

- P12,L15-21: belongs to section 4.2?

- P13, L24: NEMO-ERSEM chlorophyll is not so good really?

- P16, L6: specific issues =? please expand

- Table 2: are the question marks in the NEMO-ERSEM parameters typographic errors?

- multiple occurrences of 'less good': not wrong, but wouldn't 'not as good' be more natural?

- P24, L9: reformat Thorpe et al.

References:

Arteaga, L., Pahlow, M., Oschlies, A. Global patterns of phytoplankton nutrient and light colimitation inferred from an optimality based model. Global Biogeochem. Cycles, 28, 648-661, 2014

De Mora, L., Butenschön, M., Allen, J.I. The assessment of a global marine ecosystem model on the basis of emergent properties and ecosystem function: a case study with ERSEM. Geosci. Model Dev., 9, 59–76, 2016

---

## Referee Comment (RC3) · Anonymous Referee #3 · 26 Aug 2016

**1  Summary and overview**

In this work, HPLC data collected on two cruises in the North Sea is compared against two variants of the ERSEM model, run in different physical model environments, GETM and NEMO. After validating the SST, Chlorophyll, particulate matter, nitrate, phosphate and silicate in the two models, the size-categorised HPLC data was compared against sub-sampled model data, and discussed. The authors compared the community structure using distribution plots of total chlorophyll vs PSC fraction, then introduce a unique ternary plots style to highlight the differences between the three data sources.

The authors do not thoroughly investigate the differences in the physics or the nutritive

environment and instead focus on the functional types chlorophyll, where as these aspects of the model are crucial to understanding the origins of the divergences between these two similar models. The two versions of ERSEM are structurally very similar, and so any diverges must be due to the physical environment, the river inputs, the SPM and light models or the parameterisations. The authors have not convinced me that the differences in community structure aren't due to the physical environment or the river influx, yet they conclude that the differences are due to the sensitivity to light and nutrients parameterisation.

For instance, the GETM-ERSEM-BFM has the productivity set much higher than NEMO-ERSEM, but this appears to be compensated by much higher river nutrient fluxes than NEMO-ERSEM. This extra riverine source of nutrients leads to the stable diatom growth in the river-influenced regions. In addition, the central parts of the North Sea that do not feel the influence of the rivers have extremely low values for total chlorophyll, as the nitrate has been completely depleted there due to the high affinity of picophytoplankton. This suggests that the GETM-ERSEM-BFM community structure may be right, but for the wrong reasons. A thorough investigation of the river input, and the relationships between nutrients, chlorophyll and community structure for the two models may be needed to identify compensating errors.

Otherwise, I find that this paper to be an adequate comparison of the data and the model, but I was hoping to see more depth in the discussion and conclusions section. For instance, this paper points out that the NEMO-ERSEM simulation struggles to reproduce the August diatom concentrations seen in both the data and the other model, but does not further investigate the root cause of the differences. After providing such an in-depth description of the divergences in between the model parametrisations in section 3.3 and Tables 1 and 2 , the authors are letting themselves down by not even speculating on how specific model features could bring about the observed differences in model output. For instance, what features are in one model but not the other that could cause this? If these questions are not answerable with the current set up, perhaps the authors could propose an extension to the work that would highlight the major cause of the differences, be it nutrient affinity, nutrient stress, river influx, stratification, sinking, lysis or light susceptibility. A sensitivity study in a 1D column may be a possible extension that could answer the remaining questions about the ERSEM parametrisations.

**2   General Comments:**

The title should be revised, removing Biodiversity and "ERSEM-type models". As the authors mention in the introduction, phytoplankton community structure is an important consideration when assessing marine biodiversity, but it certainly not the only indicator. Furthermore, ERSEM-type models is not strictly defined. Please consider something like: "Observing and modelling phytoplankton community structure in the North Sea: a comparison of two biogeochemical models", but I'll let the authors come up with something better.

There is only a shallow validation of the underlying physic model validation is shown. A single figure demonstrating that the SST matches is not particularly convincing by itself, especially as one of the models already assimilates SST. In the North Sea, it is important to demonstrate that the physical model that are capable of reproducing natural vertical mixing with appropriate sources of river influx (and river nutrients). This can be done by introducing a validation of the models stratification (perhaps Mixed layer depth) and surface salinity (as a proxy for influx of fresh water). The focus of the paper should remain on the PFT's, but a demonstration of the modelled physics would allow readers to gauge how realistic of an environment the two ERSEMs live in.

The overall tone of the writing sometimes slips into a colloquial spoken-style instead of formal written tone, but this is up to the author's discretion.

I find the figure and table captions to be too brief. I would prefer having more information in them. Typically the rule of thumb is that it needs to be enough to describe the figure/table, were it taken out of context.

The text in the figures can be a little on the small side and the figures are inconsistent in terms of style, legends, axes and plotting range.

**3  Specific Comments:**

L13-L16. "A comparison of the ...is key to capturing the observed biodiversity". While I agree with this statement (after replacing "biodiversity" with "Phytoplanton Community Structure"), I'm not convinced that the work establishes it as a fact. Some changes to the discussions section are needed for it to be the case.

P7-L18: "PML-ERSEM" is used from this point onwards, but should be explicitly defined in section 3.2.

P8-L8-L9: "More explicit approach to sinking". What does this mean? As the impact of sinking SPM underpins one of the important results of the paper, it would be good to have a more in-depth description of sinking in the two models. Also, the detrital sinking rates (if they are explicitly defined), should be presented here or in Table 2.

P8-L9-L11: The light susceptibility parameters in table 2 are not directly comparable, due the differences in model choice described here. Can they instead be converted into some other measure of light susceptibility in order to make them comparable? Alternatively, please include the equations to allow readers to draw their own conclusions.

P10-L14. Does using the WOA data as validation and as a boundary conditions have an impact on the statistical independence of the validation? Similarly, figures 4, 5 and 7 have what looks like an edge effect in the GETM-ERSEM-BFM plots along the Northern Edge. The high chlorophyll growth there in figure 5 suggests that there is an influx of

nutrients from the edge condition, especially as the central region of the North sea has been completely nitrogen depleted. Dp any of the IBTS measurements sit in a region influenced by edge effects?

P11-L12: "both models were still able to capture the main observed features throughout the domain". I disagree. NEMO-ERSEM does not capture the high chlorophyll values along the German coastline, and otherwise appears relatively okay. GETM-ERSEM-BFM also does not capture the high chlorophyll in the German coastline (but does a better job than NEMO-ERSEM), but has far too little chlorophyll by a factor 10x-20x over most of the North Sea. This model also has far too much chlorophyll at the mouth of the Great Ouse river, at least a factor of 10x, perhaps up to 100x, but it's hard to tell because of the colour scale. The colour scale is hiding quite a lot of the variability in high chlorophyll values. The legend is logarithmically even between 0.1 and 5., but then adds what should be three colour bands into one (ie 5-10, 10-20,20-50 are all hidden in 5-50.) My suspicion is that using a logarithmically even colour scale for this figure will result in a change the description in the section of the paper.

P11-L25-L28 and figure 7. The WOA datasets seem a little coarse for this analysis, and are already used as boundaries conditions for both models. ICES have extensive nitrate, phosphate and silicate datasets available for the North Sea, which could be converted into a climatological dataset for future works.

P11-L30: "Both [nitrate plots] are in broad agreement with the climatology. Likewise for phosphate and silicate". I do not agree with this statement. To me, it looks like GETM-ERSEM-BFM is entirely nitrate depleted over most of the model region, by a factor of between 10x-100x. In addition, the regions around the south-eastern North Sea have far too much nitrogen (factor 10x). These regions coincide with the largest diatom concentrations, and it looks like diatom growth is strictly governed by excess or deficiency of nitrate. This suggests the presence of compensating errors, and may imply that the diatom community structure is right for the wrong reasons. Similarly for phosphate, GETM-ERSEM-BFM has significantly lower by around 10x than WOA, and

[Figure]

NEMO-ERSEM fails to capture the depleted area around the Norwegian Trench. In terms of silicate, NEMO-ERSEM has significantly depleted the silicate in the northern North Sea, which is surprising, as there are not enough diatoms in the community structure.

P12-L14: "are likely to have arisen from the differences in parametrisations." Is this true? Could some of the differences in the models also arise from differences in the physical models? For instance, if one model's mixing or riverine input results in lower surface silicate, then a difference in diatom behaviour would be expected.

P12-L31: The ternary plots to describe the three population community structure are unique to this work, and may deserve their own subsection (up to the authors discretion).

P12-L32-33: "The observations form a distinct line in this space." Surely there are some interesting findings that are a direct result of this straight line? Can it be used to inform predictions? Could you fit a predictive function to this line and compare it with the three population model?

P13-L5: Knowing about the distinct straight line in the in situ data of figure 10, would it be sensible to revise the colour scheme. How about something like Green: along the line, red: too much pico; Blue: too much nano? This would highlight the regions where the model sits along the accepted line seen the the data, and where it is wrong elsewhere.

P14-L15: Why is August a challenging month to model the in North Sea?

P14-L19-L31: Please add a clear link and some discussion about the differences in the light parametrisations of the two models in Table 2. The differences in light susceptibility between the two models may be quite significant, but its hard to tell because of the incomparable parameters. This should help support one of the major points that you make in the abstract (". . . changes in light and nutrients, is key to capturing the

observed biodiversity.")

P15-L3-L8: This is the most important part of the discussions section, and it would very informative to discuss how different model choices could have brought about the changes between these two similar simulations. (see general comments).

P15-L25: After this point, the conclusions risk becoming a little hand-wavey, non-specific and generic. For instance "close communication is needed between modellers and decision makers, so that the potential of biogeochemical models to support decision making is not lost in translation." While true, is not directly related to the conclusions of the study.

16-31 – p17-2. The closing paragraph will need to be changed to reflect the new title. "Can these two ERSEM variants simulate phytoplankton community structure? ..."

Regarding the figures, there is some inconsistency in the subplot titles and styles. Ie. Figure 2 has no sub plot titles, but figures 4,5,6,7,8 all do. The legends are in different places and have different styles. Ie Figure 5 uses discrete coloured circles, where as figure 2 uses rectangles. I prefer the larger rectangles or a color bar. (Congratulations are due for the consistent use of a single colour scheme.)

Please add a figure showing both model domains, the north sea domain used here and both cruise paths.

Figure 1: The axis labels, units, location markers are very small and hard to see. Please thicken the outline of the in situ data circles in a) and b) as in Figure 11. Also, can you please make the location makers (the capital letters) stand out more. You could do this by adding a white shadow to the letter edge or by adding an opaque text box. None of the subsequent maps include grid lines or latitude/longitude ticks, was there a reason why these plots do? If the goal is to introduce the readers to the domain, perhaps you would be better served by adding a map showing both model domains, the north sea domain used here and both cruise paths? The legend is above

the plot axes, where as in all other plots it sits on the right of the plot. Elsewhere, the chlorophyll is shown using logarithmic scale, but c) and d) use a linear scale. Parts c) and d) are described but not referenced again. What was the motivation of including two times series plots?

Figure 2: This plot has a very different style to the other plots. The sub-plots have no boarder, the coastlines are much higher resolution, they show rivers, the land is a different colour, and the coastline line is much thinner.

Figure 3: The "closest to the reference data" point of a Taylor plot is on the x-axis at avalue of 1. This point is usually marked with a dot, or the axis label "ref". Please add that point or label to this figure.

Figure 5: This figure shows the satellite Chlorophyll and both models, but it may benefit from showing the in situ data as well in another two subplots. The colour scale hides quite a lot of variability here. The legend is logarithmically even between 0.1 and 5, but then adds what should be three colour regions into one (ie 5-10, 10-20,20-50 are all hidden in 5-50.)

Figure 6: This figure and the associated description in p10-L20-23 may benefit from a logarithmic colour scale.

Figure 8: I find myself flipping back and forth between this figure, figure 2 and figure 5. There are two gaps in the 4x3 grid. Would it be possible to put total chlorophyll for each model in there? Alternatively, would it make more sense to add a second version of this figure with the three populations instead of the 4 and 5 PFTs and also the in situ data from figure 2?

Figure 9: The subplot shape is not the best here. Tall and narrow subplots seems to mask the overall shape of the data. If they were short and wide, it may make it easier to inspect the data. As mentioned above, have you considered overlaying the three population fits of of Devred 2011, Hirata 2011 or Brewin 2015 onto subfigures a) and

d) of this figure, ? This would allow a comparison of how the North Sea compares with the rest of the global ocean, as in figure 11 of Holt 2014 or in de Mora 2016. With only 85 data points you may struggle to apply the three population model to your three data sets, but it should be possible to apply the three population model to the full (not subsampled) model datasets. This could be overlayed onto subfigure b), c) e) and f). Is it even necessary to sub-sample the model data to match the in situ measurements? Judging from figure 1, the cruises have done a great job at getting an spatially even distributed data set of the North Sea. Could you comment on the detection limit of the HPLC data? The GETM-ERSEM-BFM model seems to have values much lower than that seen in data. Is that a detection limit effect not present in the model?

Figure 10: Can you please change two of the circular markers to squares and triangles, like in figure 9 so that these plots work in grey scale too?

Figure 11: This plot is the only figure in the work with a continuous colour scale, but it is the figure that would benefit the most from a discrete colour scale. In grey-scale, the legend becomes very uniform. See above comments about an alternative colour scale.

Table 1: Please add a "further reading" or "references" line with at least one reference for each model.

Table 3: In the header line, please replace ERSEM's internal PFT's names (P1-P4) with the public facing names used elsewhere in the paper.

Table 3: This table only shows total Chlorophyll, but it would be valuable to look at similar statistics for the rest of the community structure. ie pico, micro and nano in both models vs the IBTS data (but only for 2010+2011, you wouldn't need both years separately)

**4 Technical Corrections:**

Inconsistent use of "e.g." in the introduction section, especially when citing other works. My preference is to avoid it altogether. p2-L3: "e.g.", p2-L14: "e.g."., p2-L21: e.g. p2-L33 e.g. appears twice.

p2-L2: the scientific community,

p2-L5: (MICE) acronym, while clever, is never mentioned again, so is not needed.

P2-L6: remove "though"

p2-L6: ...task at hand – different scientific.... The dash ( - ) should be an em dash, a semi-colon or even a full stop.

P2-L8: colloquial: "say, global-scale...".

p2-L14: van de Molen is spelt with a lower-case v in the authors list but with a capital V elsewhere in the document and in the reference list. Is this a conscious choice? Similarly for other names: de Mora, van der Woerd, van Raaphorst, and van Leeuwen are also spelt differently in various places.

p2-L25: I don't think you need to explicitly define the "(UK)" just as you correctly don't define USA later in the same sentence.

P2-L28-L30, please re-write this sentence with more clarity.

P3-L13: replace "too" with "as well".

p6-L2,4 ,6, 25: Are these web addresses the best way to include this information? If so, please cite them instead of in-line using the Copernicus standard styles: http://publications.copernicus.org/Copernicus$_{Publications_{Reference_{Types}}}.pdf$

p3-L10, p3-L28: Would it be better to move both copies of the sentence "Details of the model configuration and forcing are given in Table 1." to the end of the introductory

paragraph at p5-L9?

P6-L22: Please reverse FOAM definition and acronym: "Forecasting Ocean Assimilation Model (FOAM)"

P7-L6-L9: Long sentence, please split into two: "reanalysis period. The biogeochemical..."

P12-L3: Please add a reminder about how the model PFTs were aggregated. Something like: "as described in section 3.4".

P14-L17: change strange wording: "confidence must be had in".

p16-L21: The dash ( - ) should be an em dash.

P16-L27: Rephrase this sentence to re-word "for confidence to be had in them".

P16-L33: replace "current" with "these two".

p34 Figure 9: Phytoplankton Size Class instead of PSC in the figure caption.

P39 Table 33 caption: Statistical comparison of log 10 (chlorophyll) against IBTS observations.

**5  Literature:**

Brewin, R. J. W., Sathyendranath, S., Hirata, T., Lavender, S. J., Barciela, R. M., and Hardman-Mountford, N. J.: A three-component model of phytoplankton size class for the Atlantic Ocean, Ecol. Model., 221, 1472–1483, doi:10.1016/j.ecolmodel.2010.02.014, 2010.

Brewin, R. J., Sathyendranath, S., Jackson, T., Barlow, R., Brotas, V., Airs, R., and Lamont, T.: Influence of light in the mixed- layer on the parameters of a three-component model of phy- toplankton size class, Remote Sens. Environ., 168, 437–
450, doi:10.1016/j.rse.2015.07.004, 2015.

de Mora, L., Butenschön, M., and Allen, J. I.: The assessment of a global marine ecosystem model on the basis of emergent properties and ecosystem function: a case study with ERSEM, Geosci. Model Dev., 9, 59-76, doi:10.5194/gmd-9-59-2016, 2016.

Devred, E., Sathyendranath, S., Stuart, V., and Platt, T.: A three component classification of phytoplankton absorption spectra: Application to ocean-color data, Remote Sens. Environ., 115, 2255–2266, doi:10.1016/j.rse.2011.04.025, 2011.

Hirata, T., Hardman-Mountford, N. J., Brewin, R. J. W., Aiken, J., Barlow, R., Suzuki, K., Isada, T., Howell, E., Hashioka, T., Noguchi-Aita, M., and Yamanaka, Y.: Synoptic relationships be- tween surface Chlorophyll-a and diagnostic pigments specific to phytoplankton functional types, Biogeosciences, 8, 311–327, doi:10.5194/bg-8-311-2011, 2011

Holt, J., Icarus Allen, J., Anderson, T. R., Brewin, R. J.W., Buten- schön, M., Harle, J., Huse, G., Lehodey, P., Lindemann, C., Memery, L., Salihoglu, B., Senina, I., and Yool, A.: Challenges in integrative approaches to modelling the marine ecosystems of the North Atlantic: Physics to fish and coasts to ocean, Prog. Oceanogr., 129, 285–313, doi:10.1016/j.pocean.2014.04.024, 2014.

---

## Short Comment (SC1) · 21 Sep 2016

**Comment to Ford et al. "Observing and modelling phytoplankton community structure in the North Sea: can ERSEM-type models simulate biodiversity?", Biogeoscience-discussion, 2016**

This work adresses an import aspect of marine ecosystem modelling and spends a commendable effort towards the assement of the North Sea phytoplankton community structure.

However, with respect to the speculation on the reason of differences in comunity structure in the two models already discussed by the referees, I'd encourage the authors to take into consideration the insights of Sinha et al. 2010, demonstrating the sensitivity of the modelled ecosystem food-web structure to the physical driver alone. These findings are potentially in contrast to the hypothesis that the causes for differences in comunity structure are mainly in the parametrisation of the biogeochemical model, which therefore requires a much more solid base (e.g. simulations using identical physical driver and external forcings).

Furthermore, it should be noted that one of the model versions used in this work (referred to as PML-ERSEM) is based on a parametrisation published in 2004. Since then, other versions and parametrisations of this ERSEM flavour have emerged (e.g.

the recent works of Butenschön et al. 2016, Ciavatta et al. 2016 and de Mora et al. 2016) that show a distinctively different comunity structure than the parametrisation used in this work (even if for different configurations, see figures 2 and 3 in de Mora et al. 2016 and figure 10 in Butenschön et al. 2016). While I wouldn't expect the authors to repeat their work with a different parametrisation, the fact that the used parametrisation (which is the established one still in use in the operational suite of the UKMO) has evolved since its publication clearly deserves mentioning as these new parametrisations are openly available in the published, peer-reviewed scientific literature.

*References:*

*Butenschön, M., Clark, J., Aldridge, J.N., Allen, J.I., Artioli, Y., Blackford, J., Bruggeman, J., Cazenave, P., Ciavatta, S., Kay, S., Lessin, G., van Leeuwen, S., van der Molen, J., de Mora, L., Polimene, L., Sailley, S., Stephens, N., Torres, R., 2016. ERSEM 15.06: a generic model for marine biogeochemistry and the ecosystem dynamics of the lower trophic levels. Geosci. Model Dev. 9, 1293–1339. doi:10.5194/gmd-9-1293-2016*

*Ciavatta, S., Kay, S., Saux-Picart, S., Butenschön, M., Allen, J.I., 2016. Decadal reanalysis of biogeochemical indicators and fluxes in the North West European shelf-sea ecosystem. J. Geophys. Res. Oceans 121, 1824–1845. doi:10.1002/2015JC011496*

*de Mora, L., Butenschön, M., Allen, J.I., 2016. The assessment of a global marine ecosystem model on the basis of emergent properties and ecosystem function: a case study with ERSEM. Geosci. Model Dev. 9, 59–76. doi:10.5194/gmd-9-59-2016*

*Sinha, B., Buitenhuis, E.T., Quéré, C.L., Anderson, T.R., 2010. Comparison of the emergent behavior of a complex ecosystem model in two ocean general circulation models. Progress in Oceanography 84, 204–224. doi:10.1016/j.pocean.2009.10.003*

---

## Author Comment (AC1) · 18 Oct 2016

We would like to thank the three anonymous reviewers and Dr Butenschön for their constructive comments. We will first give a general summary of our response, and then respond to each of the reviewers' points in turn.

The main concern raised by all the reviewers is that we did not do enough to address the impact that physical drivers could have on the differences in results, or to justify our conclusion that differences in phytoplankton community structure primarily arise from differences in biogeochemical model parameterisations. We accept this concern, and will work to expand our discussion of these points, going into more details. Furthermore, we will include some validation against in situ temperature, salinity and nutrient

data. However, we also stress that a detailed comparison of these processes was not our aim, and we will make this clearer in the manuscript. Our aim was to provide an initial assessment of phytoplankton community structure against a new set of observational data, using pre-existing model configurations, with a preliminary discussion of the reasons behind any differences. To be confident of our conclusions will require further experiments in a more controlled manner, and so we will also expand our discussion to propose targeted experiments which would lead on from our study. We will also revise the title to remove the reference to "biodiversity", add a figure showing the model domains and cruise tracks, replot other figures to be as clear, consistent and useful as possible, discuss and compare results from other studies in the literature, and address the various other points raised (detailed below).

**Detailed response to Reviewer 1**

*"I suggest using more contrasted "pole" colors in the triangular representation used in figure 11."*

This has also been raised by the other reviewers, and we will replot it with more contrasting colours.

*"As far as the models are concerned, a simple comparison between two versions of 3D-distributed ERSEM model is interesting, but rather difficult to be analysed, because both the physics and the biogeochemistry are different. As a caricatural example, the comparison between the assimilated temperature of NEMO-ERSEM and the freerunning temperature of GETM-ERSEM-BFM (figure 4) seems to me inapplicable. Secondly, when strong differences appear, the authors should try to explain more what process (physical or biogeochemical) can be thought to be possibly the main driver."*

We accept that the many differences in the physics and other experimental configurations preclude a thorough assessment of the reasons behind the differences in results. This was not our aim, and we will state this more clearly. Nonetheless, where differences in results are discussed, we will endeavour to do so in more detail. Figure 4 was

included to demonstrate inter-annual and spatial variability, as well as for validation, but we can remove it if the editor wishes.

*"I disagree a little bit with the authors when they say that "both models were still able to capture the main observed features throughout the domain.""*

We will reword this accordingly.

*"For nutrients, the too coarse maps of WOA climatologies don't allow a precise validation; but the authors could try to link the different behaviours of the two versions to their deposition and remineralisation processes : in the central North Sea, GETM-ERSEM-BFM seems to trap too much the detrital forms of N, P and Si. As far as biodiversity is concerned, GETM-ERSEM-BFM clearly gives a more realistic decomposition in phytoplanktonic functional types, while the weakness of diatoms in coastal nutrient-enriched zones seems to be strongly unrealistic in the case of NEMO-ERSEM, even in August."*

We will add some discussion regarding these points.

*"The discussion could be highly improved by replacing some general considerations by a comparison with results obtained in PFTs simulation obtained by other scientists, especially those using the DARWIN model (Follows et al., 2007)."*

We will include some discussion regarding other results in the literature.

*"the results of the two models used don't support really the optimistic answer given at the end of the paper by the authors: for me the ERSEM model has to be better calibrated before being considered as able to simulate biodiversity."*

We agree that better calibration is required; our conclusion was that such calibration appears to be achievable without requiring large fundamental changes. However, we accept this is still non-trivial, and will reword the conclusion to come across as more pragmatic.

**Detailed response to Reviewer 2**

*"1) The title is not representative: a) The title suggests this work is about biodiversity, but actually it's not. Although the 'relative abundance' of PFT's (Fig. 2,10 and 11) is an index relevant for biodiversity, the results are not at all discussed or framed in the context of biodiversity (in fact only the very last short paragraph of the manuscript refer to biodiversity again after the introduction, and replacing 'biodiversity' with 'phytoplankton community structure' would probably improve this paragraph already). b) Again, at the first glance 'ERSEM-type models' looks like more general than it is, whereas the manuscript is literally about 2 ERSEM versions, which could be more clearly indicated in the title. Accordingly, I invite co-authors to consider a more representative title (as changing the content according to the current title would basically result in a new manuscript)"*

Reviewer 3 also suggested changing the title, and we will do so accordingly. We will also replace "biodiversity" with "phytoplankton community structure" in the last paragraph as suggested.

*"2) Validation of PFT estimates is not a new endeavor: A recurring notion in the manuscript (abstract, introduction, summary and discussion) is that 'model estimates of PFTs are not validated', which is misleading. Quite a few attempts have been made previously to assess model performance with respect to reproducing the spatial distributions of PFTs, such as those making use of the data from continuous plankton recorder surveys, monitoring data obtained at stations located at different sites and estimates from satellite images, which should be acknowledged (examples are too many that pointing to specific studies would be unfair to others)."*

We will include a discussion of previous studies in the literature. Our intention was not to claim that this has never been done before, more that it is not done nearly as routinely as validation of total chlorophyll and other variables. We will check the document and reword accordingly where appropriate.

*"3) 'Observations' section is incomplete: In this section, only the procedures relating*

*to the HPLC data are provided. Other data sources, like the 'in-situ quantification of chl-a', OSTIA-SST, OC5-chlorophyll/SPM and WOA data should be introduced also in this section, and not elsewhere (e.g., as currently done in sections 4.1 and 4.2, see also the 'Technical Corrections')"*

This section was intended purely as a description of the new set of observations introduced in this study. However, we will move the description of the other data used for validation from section 4 to section 2 as requested.

*"4) 'Models' section is inadequate: a) 3.1 GETM-ERSEM-BFM: each item in the list of additional processes to 'make [ERSEM as presented by Vichi et al. (2007)] suitable for temperate shelf seas' need a reference to previous literature if possible (like in the list items iv and v), or if not, an adequate description. b) 3.2 NEMO-ERSEM: page(P)6,line(L)15-16: the employed version of NEMO appears to be not described previously (although a full description is foreseen to be provided by O'Dea at al., in prep), so here at least some hints should be provided c) 3.3 Comparison: a figure showing the complete domains of the two models would be helpful"*

We will expand the description of the models as requested, and include a figure comparing the domains. See also our response to the related comment by Reviewer 3 below.

*"5) Fig.11 is difficult to perceive: although I like the idea of overlaying measurements on the contour-plots using a ternary color map, I found it difficult to distinguish the resulting colors. Although I admit distinguishing colors is not my strength, I wonder whether using a more saturated and bright color scheme would help (eg., as in Fig.4 of Arteaga et al. 2014). The authors may also want to consider plotting the percentage histogram of each group across total chlorophyll as in Fig. 3 of De Mora et al. (2016), which does not rely at all on the color-perception abilities of the reader."*

We will replot Fig. 11 using a more contrasting colour scheme, and investigate the value of adding a histogram plot.

*"6) How to reproduce the spatial distribution of PFT's? : as mentioned above, the results suggest that the ERSEM version previously tuned more for coastal applications (GETM-ERSEM-BFM) performs better than the one used for simulating the margin of the Atlantic Ocean (NEMO-ERSEM). But considering that the performance difference wrt the PFTs basically boils down to the reproduction of the diatom blooms at the coast by GETM-ERSEM-BFM and not by NEMO-ERSEM, the obvious follow-up question is why the latter cannot reproduce the Diatom blooms at the coast, which I believe should be straightforward to figure out. It is hinted in section 3.3 and later in P15,L1-7 that the parameterization of the PFT's is likely to be the reason, but, it should be straightforward to test whether this explanation really holds. And I believe this would be a rather useful outcome of this study with a potentially wider relevance: can an open-ocean model transferred to a coastal system just by a reparameterization of the PFTs?"*

We will investigate this point further.

*"7) What drives inter-annual differences?: Figures 4-11 suggest considerable differ-ences between 2010 August and 2011 with respect to physical, chemical and bio-logical characteristics. Although these differences are acknowledged throughout the manuscript, not much is done to elaborate the mechanisms driving these differences, apart from hinting 'a set of physical drivers' to be responsible for the difference (P15, L21). Again, I feel like more could be done with the available tools and data."*

We will add more discussion to this point, and investigations suggest it may be re-lated in part to inter-annual variability in wind speeds and therefore mixed layer depths, which alters nutrient concentrations and temperatures, with knock-on effects on the ecosystem dynamics. Although as noted in our response to Reviewer 1, this is difficult to assess thoroughly given 1) that we have only one month in two years of data, 2) that the preceding months (for which we do not have PFT observations) are likely to play a role, and 3) the many differences in model configurations. Moreover, a thorough assessment was not our intention – that is left for future studies, and we will make this clearer.

[Figure]

*"Technical Corrections"*

We will address these. The question marks in Table 2 are not errors, but we accept their meaning is unclear and will modify the table accordingly. The comparison against IBTS chlorophyll was included in section 4.3 rather than 4.2 as we felt it fitted better with the rest of the validation against IBTS data.

**Detailed response to Reviewer 3**

*"The authors do not thoroughly investigate the differences in the physics or the nutritive environment and instead focus on the functional types chlorophyll, where as these aspects of the model are crucial to understanding the origins of the divergences between these two similar models. The two versions of ERSEM are structurally very similar, and so any diverges must be due to the physical environment, the river inputs, the SPM and light models or the parameterisations. The authors have not convinced me that the differences in community structure aren't due to the physical environment or the river influx, yet they conclude that the differences are due to the sensitivity to light and nutrients parameterisation."*

As noted above, given the many differences in configurations, a thorough investigation of physical and other model processes was not our aim. This study was intended as an initial assessment of whether off-the-shelf model versions can reproduce these data, with more detailed comparisons between the models left for future studies – we will make this clearer in the manuscript. That said, we will expand our discussion of the differences in results, and justify more clearly our conclusions.

*"For instance, the GETM-ERSEM-BFM has the productivity set much higher than NEMO-ERSEM, but this appears to be compensated by much higher river nutrient fluxes than NEMO-ERSEM. This extra riverine source of nutrients leads to the stable diatom growth in the river-influenced regions. In addition, the central parts of the North Sea that do not feel the influence of the rivers have extremely low values for total chlorophyll, as the nitrate has been completely depleted there due to the high affinity of*

*picophytoplankton. This suggests that the GETM-ERSEM-BFM community structure may be right, but for the wrong reasons. A thorough investigation of the river input, and the relationships between nutrients, chlorophyll and community structure for the two models may be needed to identify compensating errors."*

We agree, and will note these points in the manuscript. In particular, we will discuss two mechanisms that may be at work here (but which can't be separated): 1) the higher coastal nutrient concentrations in combination with the different nutrient affinity settings in GETM-ERSEM-BFM, allowing diatoms to out-compete other types, in contrast with identical nutrient affinity settings in NEMO-ERSEM; 2) the coincidence of diatoms with areas of high SPM concentrations in GETM-ERSEM-BFM (mostly absent in NEMO-ERSEM) in combination with greater light susceptibility of diatoms (again uniform across PFTs in NEMO-ERSEM). However, we would prefer to leave such investigations for future studies, with dedicated experiments performed in a configuration designed for such a comparison, rather than using these pre-existing runs. We will include in the discussion section a more comprehensive description of future work that should be performed as a result of this study.

*"Otherwise, I find that this paper to be an adequate comparison of the data and the model, but I was hoping to see more depth in the discussion and conclusions section. For instance, this paper points out that the NEMO-ERSEM simulation struggles to reproduce the August diatom concentrations seen in both the data and the other model, but does not further investigate the root cause of the differences. After providing such an in-depth description of the divergences in between the model parametrisations in section 3.3 and Tables 1 and 2 , the authors are letting themselves down by not even speculating on how specific model features could bring about the observed differences in model output. For instance, what features are in one model but not the other that could cause this? If these questions are not answerable with the current set up, perhaps the authors could propose an extension to the work that would highlight the major cause of the differences, be it nutrient affinity, nutrient stress,river influx, stratification,*

*sinking, lysis or light susceptibility. A sensitivity study in a 1D column may be a possible extension that could answer the remaining questions about the ERSEM parametrisations."*

We will expand the discussion and conclusions accordingly to try and address these points, and also propose further work which would answer these questions more fully. See also our comments above.

*"The title should be revised, removing Biodiversity and "ERSEM-type models". As the authors mention in the introduction, phytoplankton community structure is an important consideration when assessing marine biodiversity, but it certainly not the only indicator. Furthermore, ERSEM-type models is not strictly defined. Please consider something like: "Observing and modelling phytoplankton community structure in the North Sea: a comparison of two biogeochemical models", but I'll let the authors come up with something better."*

We will revise the title accordingly.

*"There is only a shallow validation of the underlying physic model validation is shown. A single figure demonstrating that the SST matches is not particularly convincing by itself, especially as one of the models already assimilates SST. In the North Sea, it is important to demonstrate that the physical model that are capable of reproducing natural vertical mixing with appropriate sources of river influx (and river nutrients). This can be done by introducing a validation of the models stratification (perhaps Mixed layer depth) and surface salinity (as a proxy for influx of fresh water). The focus of the paper should remain on the PFT's, but a demonstration of the modelled physics would allow readers to gauge how realistic of an environment the two ERSEMs live in."*

We will expand the validation to include some assessment against in situ observations of temperature, salinity and nutrients.

*"I find the figure and table captions to be too brief. I would prefer having more informa-*

tion in them. Typically the rule of thumb is that it needs to be enough to describe the figure/table, were it taken out of context."

We will expand these where appropriate.

*"The text in the figures can be a little on the small side and the figures are inconsistent in terms of style, legends, axes and plotting range."*

We will address these points as far as is possible.

*"L13-L16. "A comparison of the ...is key to capturing the observed biodiversity". While I agree with this statement (after replacing "biodiversity" with "Phytoplanton Community Structure"), I'm not convinced that the work establishes it as a fact. Some changes to the discussions section are needed for it to be the case."*

We will both reword this sentence and expand the discussion section.

*"P7-L18: "PML-ERSEM" is used from this point onwards, but should be explicitly defined in section 3.2."*

We will do this.

*"P8-L8-L9: "More explicit approach to sinking". What does this mean? As the impact of sinking SPM underpins one of the important results of the paper, it would be good to have a more in-depth description of sinking in the two models. Also, the detrital sinking rates (if they are explicitly defined), should be presented here or in Table 2."*

We will clarify these points.

*"P8-L9-L11: The light susceptibility parameters in table 2 are not directly comparable, due the differences in model choice described here. Can they instead be converted into some other measure of light susceptibility in order to make them comparable? Alternatively, please include the equations to allow readers to draw their own conclusions."*

We will address this.

*"P10-L14. Does using the WOA data as validation and as a boundary conditions have an impact on the statistical independence of the validation? Similarly, figures 4, 5 and 7 have what looks like an edge effect in the GETM-ERSEM-BFM plots along the Northern Edge. The high chlorophyll growth there in figure 5 suggests that there is an influx of nutrients from the edge condition, especially as the central region of the North sea has been completely nitrogen depleted. Dp any of the IBTS measurements sit in a region influenced by edge effects?"*

Yes, this will potentially affect the independence of the comparison, and we will mention this in the manuscript, although we would not expect it to have too great an influence on broad-scale patterns across the North Sea, and the WOA values inside the domain are not used by the model. There are edge effects, but the most northerly IBTS data (within the GETM-ESREM-BFM domain) are located just south of the main region of influence (see Fig. 11 for positions), so the effect on results should not be major.

*"P11-L12: "both models were still able to capture the main observed features throughout the domain". I disagree. NEMO-ERSEM does not capture the high chlorophyll values along the German coastline, and otherwise appears relatively okay. GETM-ERSEMBFM also does not capture the high chlorophyll in the German coastline (but does a better job than NEMO-ERSEM), but has far too little chlorophyll by a factor 10x-20x over most of the North Sea. This model also has far too much chlorophyll at the mouth of the Great Ouse river, at least a factor of 10x, perhaps up to 100x, but it's hard to tell because of the colour scale. The colour scale is hiding quite a lot of the variability in high chlorophyll values. The legend is logarithmically even between 0.1 and 5., but then adds what should be three colour bands into one (ie 5-10, 10-20,20-50 are all hidden in 5-50.) My suspicion is that using a logarithmically even colour scale for this figure will result in a change the description in the section of the paper."*

We will replot these figures and reword the text accordingly.

*"P11-L25-L28 and figure 7. The WOA datasets seem a little coarse for this analysis, and are already used as boundaries conditions for both models. ICES have extensive nitrate, phosphate and silicate datasets available for the North Sea, which could be converted into a climatological dataset for future works."*

We will include some validation against in situ nutrient data also collected on the IBTS cruises. As far as we are aware, the WOA datasets already make use of ICES data.

*"P11-L30: "Both [nitrate plots] are in broad agreement with the climatology. Likewise for phosphate and silicate". I do not agree with this statement. To me, it looks like GETM-ERSEM-BFM is entirely nitrate depleted over most of the model region, by a factor of between 10x-100x. In addition, the regions around the south-eastern North Sea have far too much nitrogen (factor 10x). These regions coincide with the largest diatom concentrations, and it looks like diatom growth is strictly governed by excess or deficiency of nitrate. This suggests the presence of compensating errors, and may imply that the diatom community structure is right for the wrong reasons. Similarly for phosphate, GETM-ERSEM-BFM has significantly lower by around 10x than WOA, and NEMO-ERSEM fails to capture the depleted area around the Norwegian Trench. In terms of silicate, NEMO-ERSEM has significantly depleted the silicate in the northern North Sea, which is surprising, as there are not enough diatoms in the community structure."*

We will reword the manuscript to consider these points.

*"P12-L14: "are likely to have arisen from the differences in parametrisations." Is this true? Could some of the differences in the models also arise from differences in the physical models? For instance, if one model's mixing or riverine input results in lower surface silicate, then a difference in diatom behaviour would be expected."*

This is possible, we will expand the discussion and justify further our reasoning.

*"P12-L31: The ternary plots to describe the three population community structure are*

*unique to this work, and may deserve their own subsection (up to the authors discretion)."*

We feel these would be best kept where they are, but will improve on the colours used as noted above.

*"P12-L32-33: "The observations form a distinct line in this space." Surely there are some interesting findings that are a direct result of this straight line? Can it be used to inform predictions? Could you fit a predictive function to this line and compare it with the three population model?"*

We will investigate this possibility.

*"P13-L5: Knowing about the distinct straight line in the in situ data of figure 10, would it be sensible to revise the colour scheme. How about something like Green: along the line, red: too much pico; Blue: too much nano? This would highlight the regions where the model sits along the accepted line seen the the data, and where it is wrong elsewhere."*

We will experiment with this, to see whether or not this gives clear results.

*"P14-L15: Why is August a challenging month to model the in North Sea?"*

We will expand on this in the manuscript, but getting the details of the stratification, nutrient concentrations and therefore phytoplankton concentrations is dependent on having successfully simulated processes in previous months, more so than at other times of year, as well as the processes seen during August. Also, being the height of summer, there are no strong temporal gradients driving the response of the system, like in spring and autumn, so the internal biogeochemical dynamics can play out most freely (both in reality and in the model).

*"P14-L19-L31: Please add a clear link and some discussion about the differences in the light parametrisations of the two models in Table 2. The differences in light susceptibility between the two models may be quite significant, but its hard to tell because of the*

*incomparable parameters. This should help support one of the major points that you make in the abstract (": : : changes in light and nutrients, is key to capturing the observed biodiversity.")"*

We will discuss this further in the manuscript.

*"P15-L3-L8: This is the most important part of the discussions section, and it would very informative to discuss how different model choices could have brought about the changes between these two similar simulations. (see general comments)."*

We will expand the discussion of this point.

*"P15-L25: After this point, the conclusions risk becoming a little hand-wavey, nonspecific and generic. For instance "close communication is needed between modellers and decision makers, so that the potential of biogeochemical models to support decision making is not lost in translation." While true, is not directly related to the conclusions of the study."*

We will try and tighten this section up accordingly.

*"16-31 – p17-2. The closing paragraph will need to be changed to reflect the new title. "Can these two ERSEM variants simulate phytoplankton community structure? ...""*

We will change this accordingly.

*"Regarding the figures, there is some inconsistency in the subplot titles and styles. Ie. Figure 2 has no sub plot titles, but figures 4,5,6,7,8 all do. The legends are in different places and have different styles. Ie Figure 5 uses discrete coloured circles, where as figure 2 uses rectangles. I prefer the larger rectangles or a color bar. (Congratulations are due for the consistent use of a single colour scheme.)"*

We will make these consistent.

*"Please add a figure showing both model domains, the north sea domain used here and both cruise paths."*

We will add this. The North Sea domain used for validation is the same as the GETM-ERSEM-BFM model domain, and we will make this clear in the manuscript.

*"Figure 1: The axis labels, units, location markers are very small and hard to see. Please thicken the outline of the in situ data circles in a) and b) as in Figure 11. Also, can you please make the location makers (the capital letters) stand out more. You could do this by adding a white shadow to the letter edge or by adding an opaque text box. None of the subsequent maps include grid lines or latitude/longitude ticks, was there a reason why these plots do? If the goal is to introduce the readers to the domain, perhaps you would be better served by adding a map showing both model domains, the north sea domain used here and both cruise paths? The legend is above the plot axes, where as in all other plots it sits on the right of the plot. Elsewhere, the chlorophyll is shown using logarithmic scale, but c) and d) use a linear scale. Parts c) and d) are described but not referenced again. What was the motivation of including two times series plots?"*

We will replot this figure to be easier to read and consistent with other figures. c) and d) are intended to compare the different in situ sampling methods of chlorophyll, as well as variability, and we will discuss them further.

*"Figure 2: This plot has a very different style to the other plots. The sub-plots have no boarder, the coastlines are much higher resolution, they show rivers, the land is a different colour, and the coastline line is much thinner."*

We will replot this to be consistent with the other figures.

*"Figure 3: The "closest to the reference data" point of a Taylor plot is on the x-axis at avalue of 1. This point is usually marked with a dot, or the axis label "ref". Please add that point or label to this figure."*

We will add this.

*"Figure 5: This figure shows the satellite Chlorophyll and both models, but it may benefit*

*from showing the in situ data as well in another two subplots. The colour scale hides quite a lot of variability here. The legend is logarithmically even between 0.1 and 5, but then adds what should be three colour regions into one (ie 5-10, 10-20,20-50 are all hidden in 5-50.)"*

We decided against including the in situ data as we felt that would duplicate the information in Fig. 1, and also they do not coincide in time (the satellite data being monthly composites, and the IBTS data discrete data with each data point collected at a different time). Hence it would risk sending the wrong message about the validity of the satellite data. However, we can add it if the editor wishes. We will also modify the colour scale, as with Fig. 1.

*"Figure 6: This figure and the associated description in p10-L20-23 may benefit from a logarithmic colour scale."*

We debated whether the scale should be linear or logarithmic, and settled on linear, but we will revisit that debate.

*"Figure 8: I find myself flipping back and forth between this figure, figure 2 and figure 5. There are two gaps in the 4x3 grid. Would it be possible to put total chlorophyll for each model in there? Alternatively, would it make more sense to add a second version of this figure with the three populations instead of the 4 and 5 PFTs and also the in situ data from figure 2?"*

We will investigate these different options and see what seems to work best.

*"Figure 9: The subplot shape is not the best here. Tall and narrow subplots seems to mask the overall shape of the data. If they were short and wide, it may make it easier to inspect the data. As mentioned above, have you considered overlaying the three population fits of of Devred 2011, Hirata 2011 or Brewin 2015 onto subfigures a) and d) of this figure, ? This would allow a comparison of how the North Sea compares with the rest of the global ocean, as in figure 11 of Holt 2014 or in de Mora 2016. With*

*only 85 data points you may struggle to apply the three population model to your three data sets, but it should be possible to apply the three population model to the full (not subsampled) model datasets. This could be overlaid onto subfigure b), c) e) and f). Is it even necessary to sub-sample the model data to match the in situ measurements? Judging from figure 1, the cruises have done a great job at getting an spatially even distributed data set of the North Sea. Could you comment on the detection limit of the HPLC data? The GETM-ERSEM-BFM model seems to have values much lower than that seen in data. Is that a detection limit effect not present in the model?"*

We will replot these to be short and wide, and look into including the three population fits either overlaid or as a separate subplot. We feel that sub-sampling the model data is important for a robust comparison with the observations. However, we agree that this limits the number of data points, so will look into including additional subplots using all the model grid points, and applying the three population model to these. The HPLC data has a detection limit of 0.01 mg m-3, but all observations were above this limit, hence why no cut-off was applied to the model data. We will mention this in the text.

*"Figure 10: Can you please change two of the circular markers to squares and triangles, like in figure 9 so that these plots work in grey scale too?"*

Yes, we will do this.

*"Figure 11: This plot is the only figure in the work with a continuous colour scale, but it is the figure that would benefit the most from a discrete colour scale. In grey-scale, the legend becomes very uniform. See above comments about an alternative colour scale."*

We will change this to be discrete, and also modify the colour scale, as has been raised by all reviewers.

*"Table 1: Please add a "further reading" or "references" line with at least one reference for each model."*

[Figure]

We will add this.

*"Table 2: In the header line, please replace ERSEM's internal PFT's names (P1-P4) with the public facing names used elsewhere in the paper."*

We will modify the headers, and make the parameter names clearer as raised by Reviewer 2.

*"Table 3: This table only shows total Chlorophyll, but it would be valuable to look at similar statistics for the rest of the community structure. ie pico, micro and nano in both models vs the IBTS data (but only for 2010+2011, you wouldn't need both years separately)"*

We will add these statistics.

*"Technical Corrections"*

We will address all these.

**Detailed response to M. Butenschön**

*"However, with respect to the speculation on the reason of differences in comunity structure in the two models already discussed by the referees, I'd encourage the authors to take into consideration the insights of Sinha et al. 2010, demonstrating the sensitivity of the modelled ecosystem food-web structure to the physical driver alone. These findings are potentially in contrast to the hypothesis that the causes for differences in comunity structure are mainly in the parametrisation of the biogeochemical model, which therefore requires a much more solid base (e.g. simulations using identical physical driver and external forcings)."*

As suggested by the other referees, we will expand our discussion to further consider the roles of different drivers, and will take the work of Sinha et al. (2010) into account as part of this discussion. We will also make clear that further experiments are likely required to come to firm conclusions on this issue, and will outline potential future work.

*"Furthermore, it should be noted that one of the model versions used in this work (referred to as PML-ERSEM) is based on a parametrisation published in 2004. Since then, other versions and parametrisations of this ERSEM flavour have emerged (e.g. the recent works of Butenschön et al. 2016, Ciavatta et al. 2016 and de Mora et al. 2016) that show a distinctively different comunity structure than the parametrisation used in this work (even if for different configurations, see figures 2 and 3 in de Mora et al. 2016 and figure 10 in Butenschön et al. 2016). While I wouldn't expect the authors to repeat their work with a different parametrisation, the fact that the used parametrisation (which is the established one still in use in the operational suite of the UKMO) has evolved since its publication clearly deserves mentioning as these new parametrisations are openly available in the published, peer-reviewed scientific literature."*

We will include further discussion about recent developments which have been made to PML-ERSEM, particularly within the SSB framework, and the results that have been obtained from this work. We will also outline future work that could address the points raised in our study, and which would likely best be done within the SSB modelling framework that you and others have developed.
* * *

---

## Author Response (AR1)

**Observing and modelling phytoplankton community structure in the North Sea: can ERSEM-type models simulate biodiversity?**

Response to reviewers

We would like to thank the three anonymous reviewers and Dr Butenschön for their constructive comments. We will first give a general summary of our response, and then respond to each of the reviewers' points in turn.

The main concern raised by all the reviewers is that we did not do enough to address the impact that physical drivers could have on the differences in results, or to justify our conclusion that differences in phytoplankton community structure primarily arise from differences in biogeochemical model parameterisations. We accept this concern, and have expanded our discussion of these points, going into more details. Furthermore, we have included some validation against in situ salinity and nutrient data. However, we also stress that a detailed comparison of these processes was not our aim, and have made this clearer in the manuscript. Our aim was to provide an initial assessment of phytoplankton community structure against a new set of observational data, using pre-existing model configurations, with a preliminary discussion of the reasons behind any differences. To be confident of our conclusions will require further experiments in a more controlled manner, and so we have also expanded our discussion to propose targeted experiments which would lead on from our study. We have also revised the title to remove the reference to "biodiversity", added a figure showing the model domains and added two further validation figures, replotted other figures to be as clear, consistent and useful as possible, discussed and compared results from other studies in the literature, and addressed the various other points raised (detailed below).

Detailed response to Reviewer 1

*"I suggest using more contrasted "pole" colors in the triangular representation used in figure 11."*

We have replotted this with more contrasting colours and a discrete colour scale.

*"As far as the models are concerned, a simple comparison between two versions of 3D-distributed ERSEM model is interesting, but rather difficult to be analysed, because both the physics and the biogeochemistry are different. As a caricatural example, the comparison between the assimilated temperature of NEMO-ERSEM and the freerunning temperature of GETM-ERSEM-BFM (figure 4) seems to me inapplicable. Secondly, when strong differences appear, the authors should try to explain more what process (physical or biogeochemical) can be thought to be possibly the main driver."*

We accept that the many differences in the physics and other experimental configurations preclude a thorough assessment of the reasons behind the differences in results. This was not our aim, and we have added text to Sect. 3.3 and Sect. 5 to state this more clearly. Nonetheless, where differences in results are discussed, we have endeavoured to do so in more detail, adding text to Sect. 5. Figure 4 was included to demonstrate inter-annual and spatial variability, as well as for validation, so we have kept this, and added more analysis of inter-annual variability to Sect. 5.

*"I disagree a little bit with the authors when they say that "both models were still able to capture the main observed features throughout the domain.""*

We have reworded this accordingly.

*"For nutrients, the too coarse maps of WOA climatologies don't allow a precise validation; but the authors could try to link the different behaviours of the two versions to their deposition and remineralisation processes : in the central North Sea, GETM-ERSEM-BFM seems to trap too much the detrital forms of N, P and Si. As far as biodiversity is concerned, GETM-ERSEM-BFM clearly gives a more realistic decomposition in phytoplanktonic functional types, while the weakness of diatoms in coastal nutrient-enriched zones seems to be strongly unrealistic in the case of NEMO-ERSEM, even in August."*

We have added discussion regarding these points to Sect. 5.

*"The discussion could be highly improved by replacing some general considerations by a comparison with results obtained in PFTs simulation obtained by other scientists, especially those using the DARWIN model (Follows et al., 2007)."*

We have added a paragraph of discussion regarding other results in the literature, including works based on Follows et al. (2007), to Sect. 5.

*"the results of the two models used don't support really the optimistic answer given at the end of the paper by the authors: for me the ERSEM model has to be better calibrated before being considered as able to simulate biodiversity."*

We agree that better calibration is required, our conclusion was that such calibration appears to be achievable without requiring large fundamental changes. However, we accept this is still non-trivial, and have reworded the conclusion to come across more clearly and as more pragmatic.

*"Nevertheless, I think that this paper is acceptable as a milestone one on the long way to biodiversity modelling, but only if the authors add a bibliography-based discussion and lower their optimistic evaluation of their present versions of the model."*

We have added a bibliography-based discussion, and reworded various sections, particularly in Sect. 5, to be more pragmatic in our evaluation.

Detailed response to Reviewer 2

*"1) The title is not representative:*
*a) The title suggests this work is about biodiversity, but actually it's not. Although the 'relative abundance' of PFT's (Fig. 2,10 and 11) is an index relevant for biodiversity, the results are not at all discussed or framed in the context of biodiversity (in fact only the very last short paragraph of the manuscript refer to biodiversity again after the introduction, and replacing 'biodiversity' with 'phytoplankton community structure' would probably improve this paragraph already).*
*b) Again, at the first glance 'ERSEM-type models' looks like more general than it is, whereas the manuscript is literally about 2 ERSEM versions, which could be more clearly indicated in the title. Accordingly, I invite co-authors to consider a more representative*
*title (as changing the content according to the current title would basically result in a new manuscript)"*

Reviewer 3 also suggested changing the title, and we have done so accordingly, shortening it to simply "Observing and modelling phytoplankton community structure in the North Sea", so as to remove the references to "biodiversity" and "ERSEM-type models". We have also

replaced "biodiversity" with "phytoplankton community structure" in the last paragraph as suggested.

*"2) Validation of PFT estimates is not a new endeavor:*
*A recurring notion in the manuscript (abstract, introduction, summary and discussion)*
*is that 'model estimates of PFTs are not validated', which is misleading. Quite a few*
*attempts have been made previously to assess model performance with respect to*
*reproducing the spatial distributions of PFTs, such as those making use of the data from*
*continuous plankton recorder surveys, monitoring data obtained at stations located at*
*different sites and estimates from satellite images, which should be acknowledged*
*(examples are too many that pointing to specific studies would be unfair to others)."*

Our intention was not to claim that this has never been done before, more that it is not done nearly as routinely as validation of total chlorophyll and other variables. We have reworded the abstract, added text to the introduction referencing other works, and added a paragraph to the discussions section comparing our results with other works.

*"3) 'Observations' section is incomplete:*
*In this section, only the procedures relating to the HPLC data are provided. Other*
*data sources, like the 'in-situ quantification of chl-a', OSTIA-SST, OC5-chlorophyll/SPM*
*and WOA data should be introduced also in this section, and not elsewhere (e.g., as*
*currently done in sections 4.1 and 4.2, see also the 'Technical Corrections')"*

This section was intended purely as a description of the new set of observations introduced in this study. However, we have moved the description of the other data used for validation from section 4 to section 2 as requested.

*"4) 'Models' section is inadequate:*
*a) 3.1 GETM-ERSEM-BFM: each item in the list of additional processes to 'make*
*[ERSEM as presented by Vichi et al. (2007)] suitable for temperate shelf seas' need*
*a reference to previous literature if possible (like in the list items iv and v), or if not, an*
*adequate description.*
*b) 3.2 NEMO-ERSEM: page(P)6,line(L)15-16: the employed version of NEMO appears*
*to be not described previously (although a full description is foreseen to be provided by*
*O'Dea at al., in prep), so here at least some hints should be provided*
*c) 3.3 Comparison: a figure showing the complete domains of the two models would*
*be helpful"*

We have expanded the descriptions of both the models as requested, and included a figure comparing the domains (new Fig. 1).

*"5) Fig.11 is difficult to perceive:*
*although I like the idea of overlaying measurements on the contour-plots using a*
*ternary color map, I found it difficult to distinguish the resulting colors. Although I admit*
*distinguishing colors is not my strength, I wonder whether using a more saturated and*
*bright color scheme would help (eg., as in Fig.4 of Arteaga et al. 2014). The authors*
*may also want to consider plotting the percentage histogram of each group across total*
*chlorophyll as in Fig. 3 of De Mora et al. (2016), which does not rely at all on the*
*color-perception abilities of the reader."*

We have replotted Fig. 11 (now Fig. 14) using a more contrasting colour scheme and discrete colour scale. We have also added a histogram plot (new Fig. 12) as recommended.

*"6) How to reproduce the spatial distribution of PFT's? :*
*as mentioned above, the results suggest that the ERSEM version previously tuned*

*more for coastal applications (GETM-ERSEM-BFM) performs better than the one used for simulating the margin of the Atlantic Ocean (NEMO-ERSEM). But considering that the performance difference wrt the PFTs basically boils down to the reproduction of the diatom blooms at the coast by GETM-ERSEM-BFM and not by NEMO-ERSEM, the obvious follow-up question is why the latter cannot reproduce the Diatom blooms at the coast, which I believe should be straightforward to figure out. It is hinted in section 3.3 and later in P15,L1-7 that the parameterization of the PFT's is likely to be the reason, but, it should be straightforward to test whether this explanation really holds. And I believe this would be a rather useful outcome of this study with a potentially wider relevance: can an open-ocean model transferred to a coastal system just by a reparameterization of the PFTs?"*

We have added text to Sect. 5 discussing this issue.

*"7) What drives inter-annual differences?:*
*Figures 4-11 suggest considerable differences between 2010 August and 2011 with respect to physical, chemical and biological characteristics. Although these differences are acknowledged throughout the manuscript, not much is done to elaborate the mechanisms driving these differences, apart from hinting 'a set of physical drivers' to be responsible for the difference (P15, L21). Again, I feel like more could be done with the available tools and data."*

We have added more text to Sect. 5, describing how the physical variability appears to have driven the biogeochemical variability in the models. Although as noted in our response to Reviewer 1, this is difficult to assess thoroughly given the many differences in model configurations, and a thorough assessment was not our intention – that is left for future studies, and we have added suggestions for these.

*"Technical Corrections*
*- P6, L34: sediments=? SPM?*

SPM. We have clarified this.

*- P7, L3: nutrients =? of all forms, or dissolved inorganic form only?*

Nitrate, phosphate and silicate. We have clarified this.

*- P7, L28,L30: first occurrence of the 'PML-ERSEM'. Although what this means should be clear to the careful reader, but better stick with the previous definitions.*

We have removed references to PML-ERSEM, and simply used NEMO-ERSEM.

*- P9, L11: first occurrence of MODIS (also consider my specific comment (SC)-3)*

The acronym has been expanded, and included in Sect. 2.

*- P9, L26-27: 'Picophytoplankton...' reference needed, and consider SC-3*
*- P10, L33-34: 'The largest PFT...' reference needed and consider SC-3*

We have now detailed the pigments considered in Sect. 2, with references.

*- P10, L9-14: consider SC-3*

We have moved this to Sect. 2 as requested.

*- P12,L15-21: belongs to section 4.2?*

The comparison against IBTS chlorophyll was included in section 4.3 rather than 4.2 as we felt it fitted better with the validation of PFTs, which are a function of the IBTS chlorophyll data.

*- P13, L24: NEMO-ERSEM chlorophyll is not so good really?*

We have reworded this.

*- P16, L6: specific issues =? please expand*

We have reworded this to "results from this current study should further inform future model development."

*- Table 2: are the question marks in the NEMO-ERSEM parameters typographic errors?*

The question marks in Table 2 are not errors, but we accept their meaning is unclear and have modified the table accordingly. They were intended to indicate parameters for the four model variables, we have changed this to the notation (e.g.) P1 … P4.

*- multiple occurrences of 'less good': not wrong, but wouldn't 'not as good' be more natural?*

We have changed these as suggested.

*- P24, L9: reformat Thorpe et al."*

We have done this.

Detailed response to Reviewer 3

*"The authors do not thoroughly investigate the differences in the physics or the nutritive environment and instead focus on the functional types chlorophyll, where as these aspects of the model are crucial to understanding the origins of the divergences between these two similar models. The two versions of ERSEM are structurally very similar, and so any diverges must be due to the physical environment, the river inputs, the SPM and light models or the parameterisations. The authors have not convinced me that the differences in community structure aren't due to the physical environment or the river influx, yet they conclude that the differences are due to the sensitivity to light and nutrients parameterisation."*

As noted above, given the many differences in configurations, a thorough investigation of physical and other model processes was not our aim. This study was intended as an initial assessment of whether off-the-shelf model versions can reproduce these data, with more detailed comparisons between the models left for future studies – we have made this clearer in the manuscript. That said, we have expanded our discussion of the differences in results, and justified more clearly our conclusions. See below for details. We have also added text to Sect. 5 further considering the possible role of the physics, and suggesting further experiments to investigate this.

*"For instance, the GETM-ERSEM-BFM has the productivity set much higher than NEMO-ERSEM, but this appears to be compensated by much higher river nutrient fluxes than NEMO-ERSEM. This extra riverine source of nutrients leads to the stable*

*diatom growth in the river-influenced regions. In addition, the central parts of the North Sea that do not feel the influence of the rivers have extremely low values for total chlorophyll, as the nitrate has been completely depleted there due to the high affinity of picophytoplankton. This suggests that the GETM-ERSEM-BFM community structure may be right, but for the wrong reasons. A thorough investigation of the river input, and the relationships between nutrients, chlorophyll and community structure for the two models may be needed to identify compensating errors."*

We have checked the river inputs for the GETM-ERSEM-BFM model, and found these to be correct. More generally we agree, and have added a consideration of these points to the discussion in Sect. 5. However, we would prefer to leave such investigations for future studies, with dedicated experiments performed in a configuration designed for such a comparison, rather than using these pre-existing runs. We have therefore included in the discussion section a more comprehensive description of future work that should be performed as a result of this study.

*"Otherwise, I find that this paper to be an adequate comparison of the data and the model, but I was hoping to see more depth in the discussion and conclusions section. For instance, this paper points out that the NEMO-ERSEM simulation struggles to reproduce the August diatom concentrations seen in both the data and the other model, but does not further investigate the root cause of the differences. After providing such an in-depth description of the divergences in between the model parametrisations in section 3.3 and Tables 1 and 2 , the authors are letting themselves down by not even speculating on how specific model features could bring about the observed differences in model output. For instance, what features are in one model but not the other that could cause this? If these questions are not answerable with the current set up, perhaps the authors could propose an extension to the work that would highlight the major cause of the differences, be it nutrient affinity, nutrient stress,river influx, stratification, sinking, lysis or light susceptibility. A sensitivity study in a 1D column may be a possible extension that could answer the remaining questions about the ERSEM parametrisations."*

We have added to Sect. 5 a discussion of the root causes of the inter-annual variability mentioned, as well as further discussion of the other points raised. We have also added a paragraph suggesting further experiments which would extend the work accordingly.

*"The title should be revised, removing Biodiversity and "ERSEM-type models". As the authors mention in the introduction, phytoplankton community structure is an important consideration when assessing marine biodiversity, but it certainly not the only indicator. Furthermore, ERSEM-type models is not strictly defined. Please consider something like: "Observing and modelling phytoplankton community structure in the North Sea: a comparison of two biogeochemical models", but I'll let the authors come up with something better."*

We have removed "biodiversity" and "ERSEM-type models", shortening the title to simply "Observing and modelling phytoplankton community structure in the North Sea".

*"There is only a shallow validation of the underlying physic model validation is shown. A single figure demonstrating that the SST matches is not particularly convincing by itself, especially as one of the models already assimilates SST. In the North Sea, it is important to demonstrate that the physical model that are capable of reproducing natural vertical mixing with appropriate sources of river influx (and river nutrients). This can be done by introducing a validation of the models stratification (perhaps Mixed layer depth) and surface salinity (as a proxy for influx of fresh water). The focus of the paper should remain on the PFT's, but a demonstration of the modelled physics would allow readers to gauge how realistic of an environment the two ERSEMs live in."*

We have expanded the validation to include some assessment against in situ observations of salinity and nutrients from the IBTS cruises.

*"I find the figure and table captions to be too brief. I would prefer having more information in them. Typically the rule of thumb is that it needs to be enough to describe the figure/table, were it taken out of context."*

We have expanded these where appropriate.

*"The text in the figures can be a little on the small side and the figures are inconsistent in terms of style, legends, axes and plotting range."*

We have replotted many of the figures to address these points.

*"L13-L16. "A comparison of the ...is key to capturing the observed biodiversity". While I agree with this statement (after replacing "biodiversity" with "Phytoplanton Community Structure"), I'm not convinced that the work establishes it as a fact. Some changes to the discussions section are needed for it to be the case."*

We have both reworded this sentence and expanded the discussion section.

*"P7-L18: "PML-ERSEM" is used from this point onwards, but should be explicitly defined in section 3.2."*

For clarity, we have removed reference to "PML-ERSEM", and simply used "NEMO-ERSEM".

*"P8-L8-L9: "More explicit approach to sinking". What does this mean? As the impact of sinking SPM underpins one of the important results of the paper, it would be good to have a more in-depth description of sinking in the two models. Also, the detrital sinking rates (if they are explicitly defined), should be presented here or in Table 2."*

We have added this to the description of the models in Sect. 3, and added the sinking rates to Table 2.

*"P8-L9-L11: The light susceptibility parameters in table 2 are not directly comparable, due the differences in model choice described here. Can they instead be converted into some other measure of light susceptibility in order to make them comparable? Alternatively, please include the equations to allow readers to draw their own conclusions."*

We have included in Sect. 3 a more detailed description of the light susceptibility methods of each model.

*"P10-L14. Does using the WOA data as validation and as a boundary conditions have an impact on the statistical independence of the validation? Similarly, figures 4, 5 and 7 have what looks like an edge effect in the GETM-ERSEM-BFM plots along the Northern Edge. The high chlorophyll growth there in figure 5 suggests that there is an influx of nutrients from the edge condition, especially as the central region of the North sea has been completely nitrogen depleted. Dp any of the IBTS measurements sit in a region influenced by edge effects?"*

Yes, this will potentially affect the independence of the comparison, and we have added mention of this to the manuscript. However, we would not expect it to have too great an influence on broad-scale patterns across the North Sea, and the WOA values inside the

domain are not used by the model. There are edge effects, but the most northerly IBTS data (within the GETM-ESREM-BFM domain) are located just south of the main region of influence (see e.g. Fig. 14 for positions), so the effect on results should not be major.

*"P11-L12: "both models were still able to capture the main observed features throughout the domain". I disagree. NEMO-ERSEM does not capture the high chlorophyll values along the German coastline, and otherwise appears relatively okay. GETM-ERSEMBFM also does not capture the high chlorophyll in the German coastline (but does a better job than NEMO-ERSEM), but has far too little chlorophyll by a factor 10x-20x over most of the North Sea. This model also has far too much chlorophyll at the mouth of the Great Ouse river, at least a factor of 10x, perhaps up to 100x, but it's hard to tell because of the colour scale. The colour scale is hiding quite a lot of the variability in high chlorophyll values. The legend is logarithmically even between 0.1 and 5., but then adds what should be three colour bands into one (ie 5-10, 10-20,20-50 are all hidden in 5-50.) My suspicion is that using a logarithmically even colour scale for this figure will result in a change the description in the section of the paper."*

We have replotted the figure accordingly, although the general picture remains the same. Nonetheless, we have modified this line of the text accordingly.

*"P11-L25-L28 and figure 7. The WOA datasets seem a little coarse for this analysis, and are already used as boundaries conditions for both models. ICES have extensive nitrate, phosphate and silicate datasets available for the North Sea, which could be converted into a climatological dataset for future works."*

We have included some validation against in situ nutrient data also collected on the IBTS cruises. As far as we are aware, the WOA datasets already make use of ICES data.

*"P11-L30: "Both [nitrate plots] are in broad agreement with the climatology. Likewise for phosphate and silicate". I do not agree with this statement. To me, it looks like GETM-ERSEM-BFM is entirely nitrate depleted over most of the model region, by a factor of between 10x-100x. In addition, the regions around the south-eastern North Sea have far too much nitrogen (factor 10x). These regions coincide with the largest diatom concentrations, and it looks like diatom growth is strictly governed by excess or deficiency of nitrate. This suggests the presence of compensating errors, and may imply that the diatom community structure is right for the wrong reasons. Similarly for phosphate, GETM-ERSEM-BFM has significantly lower by around 10x than WOA, and NEMO-ERSEM fails to capture the depleted area around the Norwegian Trench. In terms of silicate, NEMO-ERSEM has significantly depleted the silicate in the northern North Sea, which is surprising, as there are not enough diatoms in the community structure."*

We have reworded this statement, and added further discussion to the manuscript to consider these points.

*"P12-L14: "are likely to have arisen from the differences in parametrisations." Is this true? Could some of the differences in the models also arise from differences in the physical models? For instance, if one model's mixing or riverine input results in lower surface silicate, then a difference in diatom behaviour would be expected."*

This is possible, we have expanded the discussion and justified further our reasoning. We have also added further text to Sect. 5 to acknowledge the possible role of physics, and proposed further experiments to investigate this.

*"P12-L31: The ternary plots to describe the three population community structure are*

*unique to this work, and may deserve their own subsection (up to the authors discretion)."*

We feel these are best kept where they are, but have improved on the colours used as noted above.

*"P12-L32-33: "The observations form a distinct line in this space." Surely there are some interesting findings that are a direct result of this straight line? Can it be used to inform predictions? Could you fit a predictive function to this line and compare it with the three population model?"*

We have added text to Sect. 4.3 discussing this. Essentially it demonstrates that Nano and Pico remain very similar to each other, when Micro changes. This differs to some extent from the three population model.

*"P13-L5: Knowing about the distinct straight line in the in situ data of figure 10, would it be sensible to revise the colour scheme. How about something like Green: along the line, red: too much pico; Blue: too much nano? This would highlight the regions where the model sits along the accepted line seen the the data, and where it is wrong elsewhere."*

We have considered this, but felt the results to be less clear, particularly in terms of highlighting spatial patterns.

*"P14-L15: Why is August a challenging month to model the in North Sea?"*

We have expanded on this in the manuscript.

*"P14-L19-L31: Please add a clear link and some discussion about the differences in the light parametrisations of the two models in Table 2. The differences in light susceptibility between the two models may be quite significant, but its hard to tell because of the incomparable parameters. This should help support one of the major points that you make in the abstract (": : : changes in light and nutrients, is key to capturing the observed biodiversity.")"*

We have added further description and discussion of the light parameterisations.

*"P15-L3-L8: This is the most important part of the discussions section, and it would very informative to discuss how different model choices could have brought about the changes between these two similar simulations. (see general comments)."*

We have expanded the discussion, see specific responses above.

*"P15-L25: After this point, the conclusions risk becoming a little hand-wavey, nonspecific and generic. For instance "close communication is needed between modellers and decision makers, so that the potential of biogeochemical models to support decision making is not lost in translation." While true, is not directly related to the conclusions of the study."*

We have deleted most of this paragraph and added/reworded other discussion.

*"16-31 – p17-2. The closing paragraph will need to be changed to reflect the new title. "Can these two ERSEM variants simulate phytoplankton community structure? ...""*

We have changed this accordingly.

*"Regarding the figures, there is some inconsistency in the subplot titles and styles. Ie. Figure 2 has no sub plot titles, but figures 4,5,6,7,8 all do. The legends are in different places and have different styles. Ie Figure 5 uses discrete coloured circles, where as figure 2 uses rectangles. I prefer the larger rectangles or a color bar. (Congratulations are due for the consistent use of a single colour scheme.)"*

We have replotted many of the figures to improve consistency.

*"Please add a figure showing both model domains, the north sea domain used here and both cruise paths."*

We have added a figure showing the model domains. The North Sea domain used for validation is the same as the GETM-ERSEM-BFM model domain. The cruise observation locations can be seen in Fig. 2 (formerly Fig. 1) and other figures.

*"Figure 1: The axis labels, units, location markers are very small and hard to see. Please thicken the outline of the in situ data circles in a) and b) as in Figure 11. Also, can you please make the location makers (the capital letters) stand out more. You could do this by adding a white shadow to the letter edge or by adding an opaque text box. None of the subsequent maps include grid lines or latitude/longitude ticks, was there a reason why these plots do? If the goal is to introduce the readers to the domain, perhaps you would be better served by adding a map showing both model domains, the north sea domain used here and both cruise paths? The legend is above the plot axes, where as in all other plots it sits on the right of the plot. Elsewhere, the chlorophyll is shown using logarithmic scale, but c) and d) use a linear scale. Parts c) and d) are described but not referenced again. What was the motivation of including two times series plots?"*

We have replotted this figure accordingly. c) and d) are intended to compare the different in situ sampling methods of chlorophyll, as well as variability.

*"Figure 2: This plot has a very different style to the other plots. The sub-plots have no boarder, the coastlines are much higher resolution, they show rivers, the land is a different colour, and the coastline line is much thinner."*

We have replotted this figure accordingly.

*"Figure 3: The "closest to the reference data" point of a Taylor plot is on the x-axis at avalue of 1. This point is usually marked with a dot, or the axis label "ref". Please add that point or label to this figure."*

We have added this.

*"Figure 5: This figure shows the satellite Chlorophyll and both models, but it may benefit from showing the in situ data as well in another two subplots. The colour scale hides quite a lot of variability here. The legend is logarithmically even between 0.1 and 5, but then adds what should be three colour regions into one (ie 5-10, 10-20,20-50 are all hidden in 5-50.)"*

We have modified the colour scale. We decided against including the in situ data as we felt that would duplicate the information in Fig. 1 (now Fig. 2).

*"Figure 6: This figure and the associated description in p10-L20-23 may benefit from a logarithmic colour scale."*

We have changed the colour scale accordingly.

*"Figure 8: I find myself flipping back and forth between this figure, figure 2 and figure 5. There are two gaps in the 4x3 grid. Would it be possible to put total chlorophyll for each model in there? Alternatively, would it make more sense to add a second version of this figure with the three populations instead of the 4 and 5 PFTs and also the in situ data from figure 2?"*

We have considered these different options, and feel the current layout is the best compromise of clarity, conciseness, and lack of duplication.

*"Figure 9: The subplot shape is not the best here. Tall and narrow subplots seems to mask the overall shape of the data. If they were short and wide, it may make it easier to inspect the data. As mentioned above, have you considered overlaying the three population fits of of Devred 2011, Hirata 2011 or Brewin 2015 onto subfigures a) and d) of this figure, ? This would allow a comparison of how the North Sea compares with the rest of the global ocean, as in figure 11 of Holt 2014 or in de Mora 2016. With only 85 data points you may struggle to apply the three population model to your three data sets, but it should be possible to apply the three population model to the full (not subsampled) model datasets. This could be overlayed onto subfigure b), c) e) and f). Is it even necessary to sub-sample the model data to match the in situ measurements? Judging from figure 1, the cruises have done a great job at getting an spatially even distributed data set of the North Sea. Could you comment on the detection limit of the HPLC data? The GETM-ERSEM-BFM model seems to have values much lower than that seen in data. Is that a detection limit effect not present in the model?"*

We have replotted these to be short and wide. We feel that sub-sampling the model data is important for a robust comparison with the observations. However, we agree that this limits the number of data points, so have included an additional figure using all the model grid points, showing a 2D histogram equivalent to de Mora et al. (2016). This will make it easier to compare our results with those from other studies. The HPLC data has a detection limit of 0.01 mg m$^{-3}$, but all observations were above this limit, hence why no cut-off was applied to the model data.

*"Figure 10: Can you please change two of the circular markers to squares and triangles, like in figure 9 so that these plots work in grey scale too?"*

We have done this.

*"Figure 11: This plot is the only figure in the work with a continuous colour scale, but it is the figure that would benefit the most from a discrete colour scale. In grey-scale, the legend becomes very uniform. See above comments about an alternative colour scale."*

We have changed this to be discrete, and also modified the colour scale.

*"Table 1: Please add a "further reading" or "references" line with at least one reference for each model."*

We have added this.

*"Table 2: In the header line, please replace ERSEM's internal PFT's names (P1-P4) with the public facing names used elsewhere in the paper."*

We have modified the headers accordingly.

*Table 3: This table only shows total Chlorophyll, but it would be valuable to look at similar statistics for the rest of the community structure. ie pico, micro and nano in both models vs the IBTS data (but only for 2010+2011, you wouldn't need both years separately)"*

We have added these statistics.

*"Technical Corrections*
*Inconsistent use of "e.g." in the introduction section, especially when citing other works. My preference is to avoid it altogether. p2-L3: "e.g.", p2-L14: "e.g.".., p2-L21: e.g. p2-L33 e.g. appears twice.*
*p2-L2: the scientific community,*
*p2-L5: (MICE) acronym, while clever, is never mentioned again, so is not needed.*
*P2-L6: remove "though"*
*p2-L6: ...task at hand – different scientific.... The dash ( - ) should be an em dash, a semi-colon or even a full stop.*
*P2-L8: colloquial: "say, global-scale...".*
*p2-L14: van de Molen is spelt with a lower-case v in the authors list but with a capital V elsewhere in the document and in the reference list. Is this a conscious choice? Similarly for other names: de Mora, van der Woerd, van Raaphorst, and van Leeuwen are also spelt differently in various places.*
*p2-L25: I don't think you need to explicitly define the "(UK)" just as you correctly don't define USA later in the same sentence.*
*P2-L28-L30, please re-write this sentence with more clarity.*
*P3-L13: replace "too" with "as well".*
*p6-L2,4 ,6, 25: Are these web addresses the best way to include this information? If so, please cite them instead of in-line using the Copernicus standard styles: http://publications.copernicus.org/Copernicusₚublicationsₚₑfₑₓₑₙₑₑₜypes:pdf*
*p3-L10, p3-L28: Would it be better to move both copies of the sentence "Details of the model configuration and forcing are given in Table 1." to the end of the introductory paragraph at p5-L9?*
*P6-L22: Please reverse FOAM definition and acronym: "Forecasting Ocean Assimilation Model (FOAM)"*
*P7-L6-L9: Long sentence, please split into two: "reanalysis period. The biogeochemical..."*
*P12-L3: Please add a reminder about how the model PFTs were aggregated. Something like: "as described in section 3.4".*
*P14-L17: change strange wording: "confidence must be had in".*
*p16-L21: The dash ( - ) should be an em dash.*
*P16-L27: Rephrase this sentence to re-word "for confidence to be had in them".*
*P16-L33: replace "current" with "these two".*
*p34 Figure 9: Phytoplankton Size Class instead of PSC in the figure caption.*
*P39 Table 33 caption: Statistical comparison of log 10 (chlorophyll) against IBTS observations."*

We have addressed all these.

Detailed response to M. Butenschön

*"However, with respect to the speculation on the reason of differences in comunity structure in the two models already discussed by the referees, I'd encourage the authors to take into consideration the insights of Sinha et al. 2010, demonstrating the sensitivity of the modelled ecosystem food-web structure to the physical driver alone. These findings are potentially in contrast to the hypothesis that the causes for differences in comunity*

*structure are mainly in the parametrisation of the biogeochemical model, which therefore requires a much more solid base (e.g. simulations using identical physical driver and external forcings)."*

As suggested by the other referees, we have expanded our discussion to further consider the roles of different drivers, and have taken the work of Sinha et al. (2010) into account as part of this discussion. We have also made clear that further experiments are likely required to come to firm conclusions on this issue, and have outlined potential future work.

*"Furthermore, it should be noted that one of the model versions used in this work (referred to as PML-ERSEM) is based on a parametrisation published in 2004. Since then, other versions and parametrisations of this ERSEM flavour have emerged (e.g. the recent works of Butenschön et al. 2016, Ciavatta et al. 2016 and de Mora et al. 2016) that show a distinctively different comunity structure than the parametrisation used in this work (even if for different configurations, see figures 2 and 3 in de Mora et al. 2016 and figure 10 in Butenschön et al. 2016). While I wouldn't expect the authors to repeat their work with a different parametrisation, the fact that the used parametrisation (which is the established one still in use in the operational suite of the UKMO) has evolved since its publication clearly deserves mentioning as these new parametrisations are openly available in the published, peer-reviewed scientific literature."*

We have expanded the discussion about recent developments which have been made to ERSEM, particularly within the SSB framework, and the results that have been obtained from this work. We have also outlined future work that could address the points raised in our study, and which would likely best be done within the SSB modelling framework that you and others have developed.

[revised manuscript text omitted]